# Bringing Stability to Diffusion: Decomposing and Reducing Variance of Training Masked Diffusion Models

**Mengni Jia**[1], **Mengyu Zhou**[2], **Yihao Liu**[3], **Xiaoxi Jiang**[2], **Guanjun Jiang**[2]

[1]University of Cambridge, [2]Qwen Large Model Application Team, Alibaba, [3]Peking University

## Abstract

Masked diffusion models (MDMs) are a promising alternative to autoregressive models (ARMs), but they suffer from **inherently** much higher training variance. High variance leads to noisier gradient estimates and unstable optimization, so even pretrained MDMs and ARMs are competitive at initialization, they often diverge after task-specific training, with MDMs falling far behind. Currently, there has been no theoretical explanation or systematic solution. In this paper, we derive **the first decomposition** of MDM training variance into three sources: Ⓐ masking pattern noise, Ⓑ masking rate noise, and Ⓒ data noise – while ARMs are only affected by Ⓒ. This cleanly explains the fundamental training gap. Building on this foundation, we design six variance-reduction methods, including two core methods: (1) P-POTS, a **Pareto-optimal** $t$-sampler that minimizes training variance by sampling harder $t$ values more often with appropriately smaller update steps, and (2) MIRROR, which uses negatively correlated samples to reduce Ⓐ. Experiments show that, compared to standard MDM training, our methods improve accuracy by **7–8%** on complex reasoning tasks, while simultaneously reducing run-to-run variability to **near ARM levels**, substantially narrowing the gap with strong ARM baselines; in most settings, even the best baseline method runs remain below the worst run of our method. The code will be released at https://github.com/Qwen-Applications/StableDLLM.

## 1 Introduction

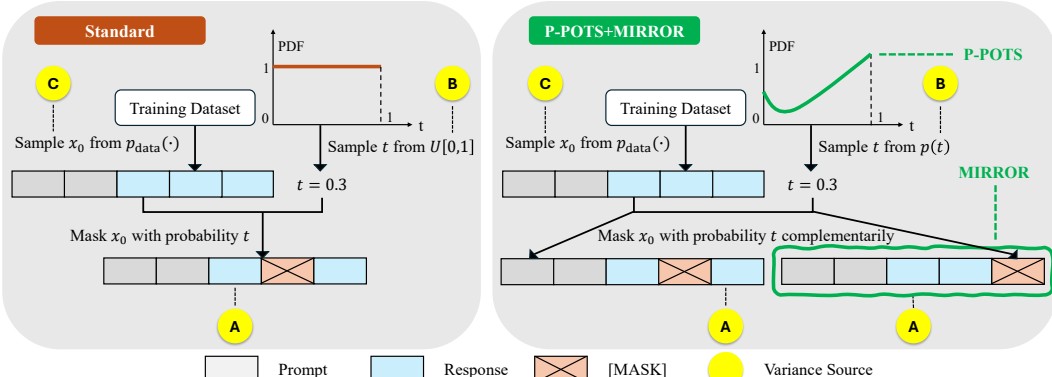

Figure 1: Graphic illustration of three sources of training variance in MDMs. The left panel illustrates standard MDM training, compared to our core methods (P-POTS and MIRROR) on the right.

Recently, masked diffusion models (MDMs) (Austin et al., 2021; Lou et al., 2023; Sahoo et al., 2024) have been viewed as a strong alternative architecture for both large language models (Dream-7B (Ye et al., 2025), LLaDA-8B (Nie et al., 2025), DiffuCoder (Gong et al., 2025)) and multi-modal models (Dimple-7B (Yu et al., 2025), MMaDA-8B (Yang et al., 2025)). They operate by masking input tokens at a randomly sampled masking rate $t$ and learning to reconstruct them. By moving

beyond sequential token modeling, they potentially address limitations of auto-regressive models (ARMs), such as lack of parallelism, exposure bias, and reversal curse (Berglund et al., 2023).

However, the noise-based training paradigm that powers MDMs also makes their training unstable. Numerous studies have reported that diffusion training suffers from high variance (Meng et al., 2021; Xu et al., 2023; Jeha et al., 2024; Kim et al., 2024; Zhu et al., 2025; Arriola et al., 2025). We refer to this phenomenon as the *training divergence* from ARMs, which manifests in two aspects:

- **Across-run variability.** Even under identical training setups, MDMs can converge to very different solutions: since parameter updates scale with loss gradients, fluctuations in losses translate directly into erratic updates. By contrast, ARMs are more predictable: with the same data order, they converge reliably to the same solution across runs.
- **Within-run sub-optimality.** Although pretrained MDMs and ARMs show comparable abilities at initialization, we consistently observe a sharp divergence after post-training: once fine-tuned on the same dataset, ARMs always outperform MDMs by wide margins, even in cases where MDMs begin from a stronger point. This highlights that, even within a single run, the variance gap makes MDMs use supervision resources less efficiently, as noisier updates slow convergence.

To address this problem, several prior works have proposed mitigation strategies. However, these efforts remain isolated and ad-hoc, lacking a unified theoretical framework for explaining the high training variance of MDMs. For example, Zhu et al. (2025) evaluates both the policy and reference ELBOs on the same time-steps and masking patterns so that their Monte Carlo errors cancel out at comparison, but their method is limited to preference optimization. Arriola et al. (2025) proposes adaptively re-fitting masking-rates samplers at each evaluation, selecting from a fixed number of candidate clipped intervals $\{[\beta_j, \omega_j]\}_{j=1}^{b} \subset [0, 1]$ to determine the best interval for training until the next evaluation. While this strategy can adjust the learning process accordingly, its reliance on such coarse candidate intervals remains heuristic. These limitations call for more principled solutions.

Motivated by this, we revisit the basic definitions to develop a systematic explanation and theoretically grounded solutions. Specifically, Sahoo et al. (2024) derived a principled training objective:

$$\mathcal{L}\text{MDM} = \mathbb{E}_{x \sim q(x),, t \sim U[0,1],, x_t \sim q(\cdot | x, t)} \left[ l_\theta(x, t, x_t) \right], \quad l_\theta = \frac{\alpha'_t}{1 - \alpha_t} \sum_{\ell=1}^{L} \log \left\langle x_\theta^\ell(x_t^{1:L}, t),\ x^\ell \right\rangle$$

(1)

for interpolating MDMs, where the loss $l_\theta$ is a function of the clean data $x$, the masking rate $t$, and the noised (masked) input $x_t$ obtained from the forward process $q(x_t \mid x, t)$. To understand the variance origins in this training objective, we derive **the first systematic variance decomposition** (proof in A.5.1). Mathematically, this yields:

$$\text{Var}_{x_0, t, x_t}(l_\theta) = \underbrace{\mathbb{E}_{x_0, t} \left[ \text{Var}_{x_t}(l_\theta \mid x_0, t) \right]}_{\text{Masking Pattern Noise Ⓐ}} + \underbrace{\mathbb{E}_{x_0} \left[ \text{Var}_t(g_\theta(x_0, t) \mid x_0) \right]}_{\text{Masking Rate Noise Ⓑ}} + \underbrace{\text{Var}_{x_0} \left( \mathbb{E}_t[g_\theta(x_0, t)] \right)}_{\text{Data Noise Ⓒ}}$$

(2)

with $g_\theta(x_0, t) = \mathbb{E}_{x_t}[l_\theta \mid x_0, t]$. This decomposition relies on **no** hard assumptions and emerges naturally, so it both explains existing methods and lays a foundation for understanding how they may complement one another. As shown in Figure 1, the MDM training variance is decomposed into three factors: Ⓐ Masking Pattern Noise (randomness from $x_t$ even if $x_0$ and $t$ are fixed), Ⓑ Masking Rate Noise (variability in the expected loss $g(x_0, t)$ across different $t$ for the same $x_0$), and Ⓒ Data Noise (variability across $x_0$, as some are inherently easier or harder to predict). In contrast, ARMs are subject to only data noise Ⓒ. Therefore, the high training variance of MDMs is explained by the extra sources Ⓐ and Ⓑ, highlighting the need to reduce them to stabilize MDM training.

To address this challenge, we introduce six techniques, including two core methods (P-POTS and MIRROR in §3.1) and four additional ones. Specifically, P-POTS is the **Pareto optimal** $t$-sampler among all unbiased ones in minimizing Ⓐ, Ⓑ and Ⓒ: it focuses training on high-variance $t$-regions while down-weighting their updates to prevent destabilizing optimization. MIRROR, on the other hand, constructs two complementary noised inputs and employs their negative correlation to explicitly reduce Ⓐ. Crucially, these two methods exhibit strong synergy, leading us to recommend P-POTS alone for cost-efficiency and P-POTS+MIRROR for maximal performance.

Experiments validate our claims. On GSM8K, the accuracy of standard MDM training ranges from 50.6% to 53.7%, while P-POTS+MIRROR improves to 58.6%–62.0%. On HiTab, the accuracy

range rises from 52.9%–62.6% to 66.0%–68.6%. On these textual datasets, our method allows MDMs to match or even surpass ARMs, with run-to-run variability reduced to near-ARM levels. On text-to-image-2M, the CLIP score range narrows from 28.61-34.28 to 34.10-35.27.

Our major contributions can be summarized as follows:

- We derive the first systematic variance decomposition for MDM training as a foundational framework (Eq. 2) and link prior w orks to relevant variance factors they actually address (§2).
- We introduce six variance-reduction methods, with P-POTS Pareto optimal and P-POTS+MIRROR giving the best results. Each method is theoretically grounded with proofs or analysis, simple to implement (requiring little or no hyperparameter tuning) and robust across datasets.
- We validate our methods on both textual and multi-modal data across various masked diffusion models, demonstrating their effectiveness and generality in stabilizing MDM training.

## 2 RELATED WORK

The inherently high training variance in diffusion models has been empirically observed. Several remedies have been proposed, yet most of them target continuous diffusion models that are widely used in image generation, and relatively few of them directly apply to masked diffusion models, the model architecture behind dLLMs (diffusion LLMs). Moreover, due to the lack of a theoretical decomposition of the variance sources, the methods remain largely isolated and heuristic. Concretely:

- Meng et al. (2021) proposes using denoising models trained at antithetic levels of Gaussian noise, which reduces mask pattern noise (A), but is limited to continuous diffusion models.
- Xu et al. (2023) computes the denoising score as a self-normalized importance-sampled weighted average over a reference batch. It reduces (A) and (B), but introduces systematic bias with finite reference batches and suffer weight degeneracy in high-dimensional or poorly covered posteriors.
- Jeha et al. (2024) uses a $k$th order Taylor expansion of the denoising score as a control variate and subtracting it (with an optimally fitted coefficient) from the Monte Carlo estimate. However, it depends on closed-form Gaussian moment calculations that don't generalize to non-Gaussian noise (e.g., masked diffusion models). Moreover, the method's reliance on low-order Taylor approximations leads to ineffective variance reduction under high noise or rapidly varying derivatives.
- Kim et al. (2024) dynamically adjusts the probability of sampling each diffusion timestep based on online estimates of gradient variance. This reduces mask ratio noise (B), but does not apply importance reweighting, which biases their loss estimates; moreover, the sampler is restricted to a simple $\text{Beta}(a, b)$, which may fail to capture the true optimal sampling distribution.
- Zhu et al. (2025) formalizes the variance of ELBO-based gradient estimates in masked diffusion language models and employs optimal Monte Carlo budget allocation and antithetic sampling, reducing the mask pattern noise (A), but is limited in preference optimization.
- Arriola et al. (2025) select a $t$-sampling interval by evaluating candidate sub-intervals and choosing the one that minimizes evaluation variance, which reduces (A)+(B)+(C). While they justify clipping away extreme $t$s (e.g., to $[0.45, 0.95]$), our analysis suggests extremes can be worth emphasizing, so their gains likely come from clipping that sometimes approximates the optimal $p(t)$.

## 3 METHOD

Let's recap the standard training algorithm for masked diffusion models (MDMs). Let $x_0$ denote the concatenation of prompt and response tokens and $P$ denote the number of response tokens, then:

1. Sample clean data $x_0$ from the data distribution.
2. Sample a masking rate $t \sim \mathcal{U}[0, 1]$.
3. For each *eligible* token $i$, sample $U_i \sim \mathcal{U}[0, 1]$, and construct $x_t$ by replacing $x_0(i)$ with [MASK] if $U_i < t$, so each *eligible* token is independently masked with probability $t$.
4. Compute the sample loss $l_\theta(x_0, t, x_t) = -\frac{1}{Pt} \sum_{i=1}^{P} \mathbb{1}[x_t(i) = \text{[MASK]}] \log p_\theta(x_0(i) \mid x_t)$.

5. Average $l_\theta(x_0, t, x_t)$ across the batch to obtain $\hat{l}_\theta$ and update $\theta$ via backpropagation.

Which tokens are *eligible* for masking depends on the training stage: in pretraining both prompts and responses are *eligible*, while in supervised fine-tuning only responses are. This algorithm is a special case of discrete diffusion (Austin et al., 2021) – MDMs with $\alpha_t = t$ (see Eq. 1). Nearly all pretrained MDMs adopt this setting. Beyond this, MDMs can model both text and image tokens (Yang et al., 2025), making it a unified framework for both language and multi-modal modeling.

To minimize the training objective in Eq. 1, gradient-based methods such as stochastic gradient descent (SGD) are typically used (Robbins & Monro, 1951). However, these methods require an empirical estimate of Eq. 1, which is usually the batch mean of losses $\frac{1}{B} \sum_{i=1}^{B} \ell(x_0^{(i)}, t_i, x_t^{(i)} \mid \theta)$. For ARMs, this estimator is generally reliable, since the variance (Eq. 2) arises only from Ⓒ. However, in MDM training, we question its reliability, since extra sources Ⓐ and Ⓑ come into play. While the standard MDM training procedure gives an unbiased estimator of Eq. 1, its variance can be significantly high. This raises a central question – can we construct an alternative estimator that remains unbiased but achieves lower variance, providing a more stable basis for optimization?

In the rest of this section, we introduce six methods – two core methods and then four others – to address the problem. A qualitative analysis of their combinations is provided in A.6.

## 3.1 CORE METHODS

### 3.1.1 P-POTS: PARAMETRIC-PARETO OPTIMAL t-SAMPLING

**P-POTS** (**P**arametric-**P**areto **O**ptimal **T**imestep **S**ampling) replaces the masking rate sampler $t \sim U[0, 1]$ with a data-fitted non-uniform one to reduce Ⓐ+Ⓑ+Ⓒ. To estimate Eq. 1, standard training uses $t \sim U[0, 1]$ directly: unbiased but high variance (§3). Instead, one could use another $t$-sampler that remains unbiased but reduces variance via importance weights (Kahn & Marshall, 1953).

We now derive the optimal unbiased $t$-sampler in the view of minimizing Ⓐ+Ⓑ+Ⓒ. If we instead sample $t \sim p(t)$, we can re-weight to form $\frac{1}{p(t)} l_\theta$. This estimator has the same expectation as $t \sim U[0, 1]$ and is thus still unbiased in estimating the training objective 1, as shown below:

$$\mathbb{E}_{t \sim p, x_t}\left[\frac{1}{p(t)} l_\theta(x_0, t, x_t)\right] = \int_0^1 p(t) \frac{1}{p(t)} g(t) dt = \int_0^1 g(t)\, dt = \mathbb{E}_{t \sim U, x_t}(l_\theta), \qquad (3)$$

With Eq. 3 establishing the unbiasedness, we next turn to minimizing its variance from $x_0, t$ and $x_t$, which is exactly Ⓐ+Ⓑ+Ⓒ in Eq. 2. Define $v(t) = \text{Var}_{x_0, x_t}(l_\theta \mid t)$ and $g(t) = E_{x_0, x_t}(l_\theta \mid t)$. Under regularity conditions, Fubini's theorem (Rudin, 1987) allows us to change the order of expectation between $t$ and $x_0, x_t$. The variance can then be written as (detailed proof in A.5.2):

$$Ⓐ + Ⓑ + Ⓒ = \text{Var}_{x_0, t, x_t}(\frac{1}{p(t)} l_\theta(x_0, t, x_t))) = \int_0^1 \frac{g(t)^2 + v(t)}{p(t)}\, dt - \left(\int_0^1 g(t)\, dt\right)^2. \quad (4)$$

To minimize Eq. 4 over all $t$-samplers, one can use Lagrange multipliers (Lagrange, 1811), yielding the optimal choice $p^*(t) = \frac{\sqrt{g(t)^2 + v(t)}}{\int_0^1 \sqrt{g(t)^2 + v(t)}\, dt} \propto \sqrt{g(t)^2 + v(t)}$. In minimizing Ⓐ, Ⓑ and Ⓒ, this is *Pareto-optimal* among all unbiased $t$-samplers of $U[0, 1]$: no other choice improves them jointly.

**Algorithm of P-POTS.** Since $g(t)$ and $v(t)$ are unknown, P-POTS estimates them empirically: before training, for each of $\{x_0^{(i)}\}_{i=1}^a$, draw $b$ values $\{t_j\}_{j=1}^b$ and, for each $(x_0^{(i)}, t_j)$, draw $c$ masked samples $\{x_t^{(i,j,k)}\}_{j=1}^c$ with losses $\ell_{i,j,k} := l_\theta(x_0^{(i)}, t_j, x_t^{(i,j,k)})$. We then compute

$$\hat{g}_j = \frac{1}{ac} \sum_{i=1}^a \sum_{k=1}^c \ell_{i,j,k}, \ \hat{v}_j = \frac{1}{a} \sum_{i=1}^a \frac{1}{c-1} \sum_{k=1}^c (\ell_{i,j,k} - \frac{1}{c} \sum_{k'=1}^c \ell_{i,j,k'})^2, \ \hat{p}_j = \sqrt{\hat{g}_j^2 + \hat{v}_j} / \sum_{j=1}^c \sqrt{\hat{g}_j^2 + \hat{v}_j}$$

for $j = 1, \ldots, b$. The weights $\{\hat{p}_j\}_{j=1}^b$ represent an empirical scatter of the underlying optimal $p^*(t)$. To capture its structure, we design a model called **EPR** (**E**xponential-**P**olynomial **R**oot) as:

$$p^{\text{EPR}}(t) = \sqrt{at^r + b(1 - t)^q + A^2 \exp(2\kappa t^m)}, \qquad a, b, A, \kappa > 0, r, q \geq 0, m > 1 \qquad (5)$$

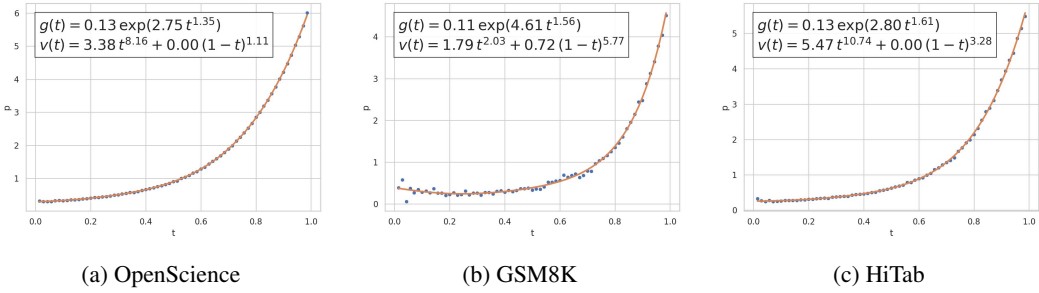

(a) OpenScience          (b) GSM8K          (c) HiTab

Figure 2: Empirical $\{p_j\}_{j=1}^b$ (scatter) and the fitted curve $p^*(t) = \sqrt{g(t)^2 + v(t)}$ (line) on three datasets: (a) OpenScience, (b) GSM8K, and (c) HiTab. The equations in each subplot show the fitted forms of $g(t)$ and $v(t)$, which together characterize the sampling distribution across masking rates $t$.

with $g^{\mathrm{EPR}}((t) = A\exp(\kappa t^m)$ and $v^{\mathrm{EPR}}((t) = b(1-t)^q + at^r$, and we fit it via KL divergence. By the EPV rule that suggests 10 data points per fitted parameter (Peduzzi et al., 1996), we set $b = 70$ and slightly raise $a$ and $c$ to 15, which proved reliable. As shown in Figure 2, EPR fits $\{\hat{p}_j\}_{j=1}^{70}$ almost perfectly with only seven parameters, suggesting it captures the characteristic form of optimal $p^*(t)$.

**Intuition behind P-POTS.** We call a $t$ *hard* to train if $\sqrt{g^2(t) + v(t)}$ is large, typically at higher $t$ (Figure 2). Our P-POTS allocates more samples to these regions, providing extra training on them, while the reweighting factor $1/p^*(t)$ keeps their contribution small so that their noisy signals do not dominate and hurt global optimization. In this way, $p^*(t)$ acts as an adaptive sampler, directing training effort where it is most needed and using resources more efficiently.

**Intuition behind EPR.** To motivate our forms of $v^{\mathrm{EPR}}(t)$ and $g^{\mathrm{EPR}}(t)$ in Eq. 5, we consider loss behavior at different $t$. The mean loss $g(t)$ reflects average task difficulty at $t$. Its growth is not just because more tokens are masked, but because entire reasoning chains are broken. Once a chain is cut, the model must guess at multiple points, and errors compound multiplicatively (or equivalently, add in log space), motivating an exponential form with delayed ignition. The design of $m > 1$ ensures $g^{\mathrm{EPR}'}(0) = 0$, since the average loss $g(t)$ at $t = 0$ is 0 and must be a local minimum.

Meanwhile, empirically, we find that $v(t)$ falls when $t$ is small, rises after a local minimum, and falls again after a local maximum – thus exhibiting up to two turning points. We explain this behavior as follows. At small $t$, a loss spike occurs if a critical set of $r$ key tokens is masked simultaneously ($\sim t^r$). At large $t$, loss remains high unless a set of $q$ key tokens survives ($\sim (1-t)^q$). Since the variance of a rare event is approximately its probability, and multiple nearly independent rare events contribute additively, these mechanisms together generate the observed curvature . Based on this explanation, we propose the model $v(t) = at^r + b(1-t)^q$.

While one may wonder if it suffices to naively fit $\{p_j\}_{j=1}^b$ with polynomials, we argue this is problematic for three reasons. First, as discussed in the next paragraph, $p(t)$ faces a drift problem: its shape may evolve during training. Although this drift is typically slow, it can become severe if the training loss remains high (*e.g.*, MMaDA on text-to-image-2M). Consequently, polynomials quickly lose effectiveness if the true functional form of $p^*(t)$ is not well understood. Experiments in A.8 show that EPR performs better in this case. Second, polynomials require manual tuning of degree, while P-POTS requires *no* tuning at all. Third, they lack interpretability and ignore known structure.

**Drift of** $p(t)$. While $p(t)$ reduces variance, it may become outdated as model evolves during training—a phenomenon we refer to as the *drift of* $p(t)$. In practice, P-POTS only fits $p^*(t)$ once before training and then replaces the uniform sampler. This incurs negligible overhead, but leaves the sampler static and therefore potentially exposed to drift. Empirically, however, the drift tends to be slow, since our experiments show strong gains even without correction; see §4. An adaptive strategy, such as periodically re-estimating $p(t)$ to re-estimate $p(t)$, could address this. Given the strong empirical performance of P-POTS, we leave such validation to future work.

### 3.1.2 MIRROR: VARIANCE REDUCTION WITH MIRRORED MASKS

**MIRROR** (Variance reduction with **MIRROR**ed masks) targets Ⓐ by creating two masked samples complementarily from the *same* $x_0$ and $t$. With the notation introduced in §3, it works as follows:

1. Sample clean data $x_0$ from the data distribution.
2. Sample a masking rate $t \sim \mathcal{U}[0, 1]$.
3. After sampling $x_0$ and $t$, for each *eligible* token $i$, sample, and generate two noisy samples:
   - $x_t^1$ by masking $x_0(i)$ if $U_i < t$
   - $x_t^2$ by masking $x_0(i)$ if $U_i > 1 - t$
4. Compute $l_j = -\frac{1}{t} \sum_{i=1}^{P} \mathbb{1}\big[x_t^j(i) = \texttt{[MASK]}\big] \log p_\theta(x_0(i) \mid x_t^j)$ for $j \in \{1, 2\}$
5. Average $\bar{l} = \frac{1}{2}(l_1 + l_2)$ across batch for backpropagation.

Compared to standard training that averages $l_1$ from a single $x_t^1$ across the batch, MIRROR uses $\bar{l}$, which remains unbiased in estimating the the training objective (Eq. 1). It uses the same number of $x_0$ samples, thereby avoiding overfitting. While it doubles training cost due to an extra forward pass, it is particularly effective on long-response datasets and shows strong synergy with P-POTS.

**Intuition behind MIRROR.** MIRROR can be understood as a hedging strategy between two assets. To reduce Ⓐ, one needs a loss estimate from $x_0$ with lower variance that arises solely from masking patterns. Suppose the token sequence is masked at $n$ positions, if the easier tokens are masked (*e.g.*, uninformative prefix tokens) versus the harder ones (*e.g.*, answer tokens), the resulting losses can differ significantly. To reduce this variance, MIRROR samples two noised inputs as described above. Due to the "complementary" masking pattern design, the two noised samples tend to yield negatively correlated losses. Consequently, regardless of whether $x_t^1$ correspond to an easier or harder case, $x_t^2$ provides the complementary one, and the averaged loss from them still remains a reliable estimate.

Ⓐ **Reduced by at Least a Half.** Given $x_0$ and $t$, the only randomness comes from $x_t$. Since $l_1$ and $l_2$ are identically distributed, define $\sigma^2 = \text{Var}(l_1) = \text{Var}(l_2)$ and $\rho = \text{Corr}(l_1, l_2) \leq 0$. Then

$$\text{Var}(\bar{l}) = \text{Var}\Big(\tfrac{1}{2}(l_1 + l_2)\Big) = \frac{1}{4}\Big(\text{Var}(l_1) + \text{Var}(l_2) + 2\,\text{Cov}(l_1, l_2)\Big) = \frac{2\sigma^2}{4}\big(1 + \rho\big) = \frac{\sigma^2}{2}\big(1 + \rho\big),$$

The effectiveness of variance reduction depends on $\rho = \text{Corr}(l_1, l_2)$. Since $\rho \in [-1, 1]$, $\text{Var}(\bar{l}) \leq \text{Var}(l_1)$, so MIRROR is never worse than standard training. Because $x_t^1$ and $x_t^2$ are constructed complementarily, $\rho$ tends to be negative, which already reduces Ⓐ by at least a half. For $t < 0.5$, the two masks have no overlap, producing even stronger negative correlation between $l_1$ and $l_2$ and thus greater variance reduction. This explains its synergy with P-POTS (discussed later in §3.1.3).

**Comparison to MultiSample-$k$.** We refer to as MultiSample-$k$ the method that estimates $l_\theta(\cdot \mid x_0, t)$ by drawing $k$ independent masked samples $x_t^{(1)}, \dots, x_t^{(k)}$ from the same masking strategy $U_i < t$ and averaging $\tilde{l} = \frac{1}{k} \sum_{j=1}^{k} l_\theta(x_t^{(j)} \mid x_0, t)$. Our comparison focuses on:

- Negative Correlation: MultiSample-2 uses independent masks, so $\text{Cov}(l_1, l_2) = 0$. MIRROR introduces natural complementarity, leading to $\text{Cov}(l_1, l_2) < 0$ and better variance reduction.
- Token Coverage: Only masked tokens contribute to gradient updates. Intuitively, masking more tokens reduces pattern noise, as each token carries less individual weight when more contribute. Standard training masks each token with probability $t$, giving expected coverage $tP$. MIRROR uses complementary masks, which together *maximize* two-view coverage as $\Pr(i \in M_1 \cup M_2) = \Pr(U_i < t) + \Pr(U_i > 1 - t) = \min(1, 2t)$. By contrast, MultiSample2 does *not* have double coverage: $\Pr(i \in M_1 \cup M_2) = 1 - \Pr(i \notin M_1 \cap M_2) = 1 - (1 - t)^2 = 2t - t^2 \leq 2t$.

### 3.1.3 SYNERGY OF P-POTS AND MIRROR

P-POTS and MIRROR are compatible because they rely on non-interacting assumptions – accurate modeling of $p(t)$ and negative correlation between $(x_t^1, x_t^2)$. However, why their combination yields synergies beyond simple additivity $(1 + 1 > 2)$ worth further investigation.

To answer this question, we return to Eq. 4, the variance in estimating the training objective ($l_\theta$ in Eq. 1). With $p(t) \propto \sqrt{g^2(t) + v(t)}$, it becomes $\left(\int_0^1 \sqrt{g^{*2}(t) + v^*(t)}\, dt\right)^2 - \left(\int_0^1 g^*(t)\, dt\right)^2$. Since this value no longer depends on $p(t)$, for fixed $g(t)$ and $v(t)$, it cannot be further reduced by altering $p(t)$. However, it can still be lowered by changing $g(t)$ and $v(t)$ during training.

Suppose $v(t) \neq 0$ and $\int_0^1 v(t)\,dt = V > 0$ is fixed. The variance depends on how close $\int_0^1 \sqrt{g^2(t) + v(t)}\,dt$ and $\int_0^1 g(t)\,dt$ are. To bring them closer, $v(t)$ should be concentrated where $g(t)$ is large, as $\sqrt{g^2(t) + v(t)}$ is less sensitive to $v(t)$ in this regime. MIRROR helps here by suppressing $v(t)$ most strongly at moderate $t$, leaving $v(t)$ relatively larger where $g(t)$ is high. Thus, MIRROR naturally shifts $v(t)$ towards the direction favored by P-POTS. Together they reinforce each other: MIRROR clears moderate-$t$ regions, while P-POTS emphasizes low- and high-$t$. This synergy is remarkable, as naive method combinations rarely deliver such gains (A.6).

## 3.2 OTHER TECHNIQUES

We introduce several other variance-reduction techniques as **baseline methods** in our experiments.

### 3.2.1 ISAD: IMPORTANCE SAMPLING ON ANSWER DELIMITERS

**ISAD** shift masking toward answer delimiters, targeting Ⓐ, by the sampling distribution:

$$q_j(t) = \begin{cases} t, & \text{if token } j \notin \text{rare ids,} \\ \min(1,\, t + \Delta), & \text{if token } j \in \text{rare ids,} \end{cases}$$

with $\Delta \in (0, 1)$. To keep the estimator unbiased, token-level losses are reweighted by $1/q_j(t)$. The intuition is to guide the model on when to output answer tokens for better downstream performance.

### 3.2.2 SYRM: SYNTAX-AND-RESPONSE MASK

**SyRM** is designed for structured data such as HTML tables, source code, or graphs. We partition tokens into two groups: (i) response tokens $R$, and (ii) syntax tokens $C$ in the prompt. For example, in an HTML table, $C$ includes tags such as `<table>`, `<tr>`, `<th>`, and `<td>`. SyRM modifies the eligibility set for masking to $R \cup C$, so that both response tokens and syntax tokens may be masked.

**Rationale.** Assuming (i) tokens within each group are homogeneous in first and second moments of token-level losses, (ii) syntax tokens are predicted more stably than response tokens, and (iii) reasoning errors couple response tokens more strongly, we show that SyRM reduces masking-pattern noise Ⓐ at the cost of a small, bounded bias (optimum shift). Overall, the mean-squared error of estimating training loss can still decrease. Intuitively, by treating easy-to-predict syntax tokens as auxiliary targets, SyRM stabilizes training and mitigates mask-pattern noise, while still preserving the primary learning signal on response tokens (see A.5.4 for the rigorous proof).

### 3.2.3 STRATS: STRATIFIED t-SAMPLING

Instead of drawing i.i.d. $t_i \sim \mathcal{U}[0, 1]$ for batch size $n$, partition $[0, 1]$ into $k$ equal strata $[j/k,\, (j+1)/k]$ with $j = 0, \dots, k-1$ and draw $\left\lfloor \frac{n}{k} \right\rfloor$ values of $t_i$ per stratum, with leftovers assigned randomly.

This approach was first introduced in Sahoo et al. (2024) with $k = n$. In this work, we prove that stratified sampling yields an unbiased estimator and reduces Ⓑ via inter-stratum variance. Furthermore, we recommend $k = \lceil \sqrt{n} \rceil$ instead of $k = n$ as a general strategy (see Appendix A.5.5). Although **StraTS** uses stratified sampling for the $t$-sampler instead of $p^*(t)$ (§3.1.1) and is not optimal for minimizing $A_{x_0} + B_{x_0}$, it still outperforms baselines and is easy to implement.

### 3.2.4 EMA: BIN-WISE EMA CONTROL VARIATE

To reduce Ⓑ, we reduce its variance arising from heteroscedastic losses across masking rates by maintaining an exponential moving average (**EMA**) of losses within bins of $t \in [0, 1]$. Each bin tracks a baseline that is updated online, yielding a piecewise-constant approximation of the expected loss against $t$. By using these bin-wise estimates as control variates, we cancel baseline-related noise more effectively than with fixed scaling, while preserving unbiasedness of gradient estimates.

This approach requires choosing the number of bins $m$ and EMA rate $\eta$, which balances stability and adaptivity. From our MSE-minimization analysis in A.5.6, we derive $m \times$ batch size per GPU $\approx 0.1 \times n \times$ (train data size), which serves as a practical rule for selecting $m$ and $\eta$.

# 4 EXPERIMENTS

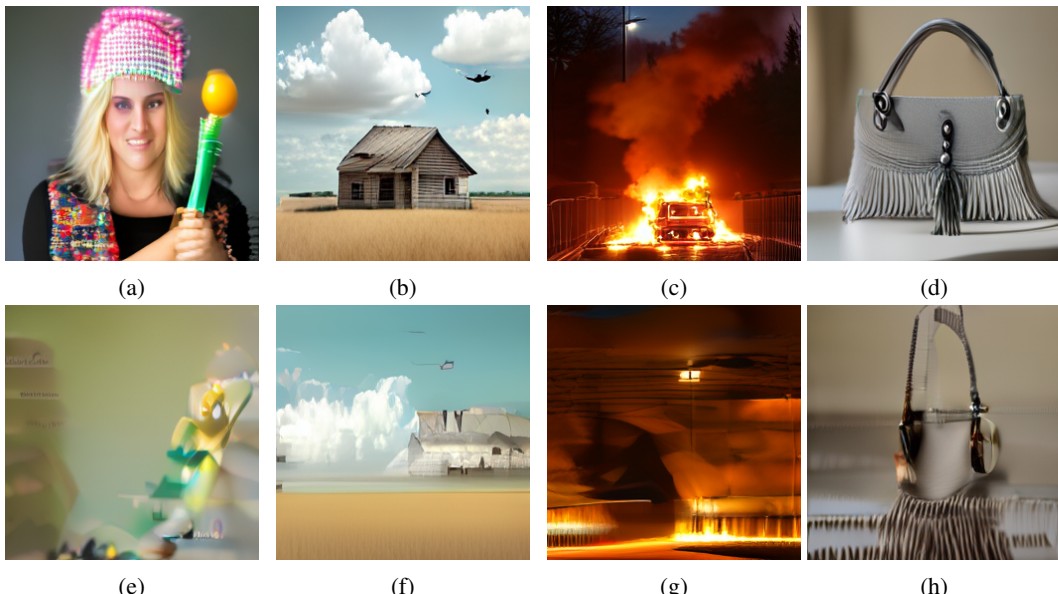

Figure 3: Images generated by MMaDA-8B-MixCoT trained with P-POTS+MIRROR (top row) and the standard method (bottom row) under the same seed. The columns correspond to the prompts: 1) A woman with blonde hair is wearing a colorful crochet headband and a black top with a floral patterned shawl. She is holding a toy sword with a green handle and a yellow ball on top; 2) A rustic wooden farmhouse with a weathered roof and a small porch stands in a field of dry grass. The sky is filled with fluffy white clouds, and a single bird is flying in the distance. The landscape is devoid of any other structures or signs of life, giving the scene a serene and isolated feel; 3) A car is on fire under a bridge with smoke billowing out. The fire is intense, with flames visible on the car's underside. The bridge has a metal railing and is located near a forested area. There is a street lamp above the bridge, and the scene appears to be at night or during dusk; 4) A close-up of a grey fringed purse with a tassel detail on the front. The purse is placed on a surface with a soft, blurred background; more case studies can be found in A.11.

Limited by resources, and given that current MDMs perform comparably to strong ARM baselines (Nie et al., 2025; Yang et al., 2025), we focus our validation on supervised fine-tuning, though our methods naturally extend to pretraining. We evaluate our methods using the setup below:

**Setting.** We train each model three times independently using seeds 42, 731 and 20231, holding model, data and all hyper-parameters fixed, and evaluate the trained models by greedy decoding. *This setting ensures difference across runs arise solely from training, not inference.*

**Tasks.** We evaluate on *three textual datasets* – OpenScience (NVIDIA Corporation, 2025), GSM8K (Cobbe et al., 2021), and HiTab (Cheng et al., 2022) – as well as one *image generation dataset*, text-to-image-2M (Hate, 2024). OpenScience is a multi-domain, knowledge-intensive dataset spanning humanities, law, and STEM; GSM8K targets mathematical reasoning; and HiTab focuses on tabular question answering. Together with the multimodal text-to-image-2M benchmark, diverse data modalities and domains are covered. Details of data processing and training/inference hyper-parameters are provided in Appendix A.7. We report pass@1 accuracy for OpenScience, GSM8K and HiTab, and CLIP score for text-to-image-2M.

**Models.** We evaluate two MDMs in total. On textual datasets, we evaluate one MDM, LLaDA-8B-Instruct (Nie et al., 2025), with three ARMs, Qwen2.5-7B-Instruct (Team, 2024), Qwen3-8B (Team, 2025) and Llama3.1-8B-Instruct (Meta AI, 2024b). For text-to-image-2M, we evaluate one multimodal MDM, MMaDA-8B-MixCoT (Yang et al., 2025).

**Baselines.** We compare our methods to two baselines: (i) standard training (§3), and (ii) the clipped noise schedule of Arriola et al. (2025), which has been reported to achieve the best performance on perplexity across most standard schedules (Chang et al., 2022), including linear, logarithmic, square root, square, and cosine. We therefore adopt it as a representative baseline.

Table 1: Per-seed performance comparison of MDMs and ARMs. The best performance within MDMs training methods is bolded. Training is repeated under three seeds, with model, data and all hyper-parameters fixed, and we evaluate trained models by greedy decoding, so difference across runs are solely from **training** rather than **inference**. Avg is the mean over available runs.

| Method (Seed) | OpenScience | | | | GSM8K | | | | HiTab | | | |
|---|---|---|---|---|---|---|---|---|---|---|---|---|
| | 42 | 731 | 20231 | Avg | 42 | 731 | 20231 | Avg | 42 | 731 | 20231 | Avg |
| LLaDA-8B-Instruct | | | | | | | | | | | | |
| P-POTS+Mirror | **50.80** | **53.80** | **53.00** | **52.53** | **62.02** | **58.61** | **60.96** | **60.53** | **66.02** | **66.67** | **68.62** | **67.10** |
| P-POTS | 45.34 | 47.60 | 47.47 | 46.80 | 59.59 | 59.36 | 56.79 | 58.58 | 60.56 | 64.74 | 58.80 | 61.37 |
| MIRROR | 45.63 | 47.25 | 46.25 | 46.38 | 56.56 | 51.40 | 53.15 | 53.70 | 64.72 | 65.04 | 63.68 | 64.48 |
| ISAD | 45.40 | – | – | 45.40 | 55.27 | – | – | 55.27 | 64.39 | – | – | 64.39 |
| SyRM | – | – | – | – | – | – | – | – | 59.79 | – | – | 59.79 |
| StraTS | 44.69 | – | – | 44.69 | 55.57 | – | – | 55.57 | 61.86 | – | – | 61.86 |
| EMA | 47.08 | – | – | 47.08 | 52.39 | – | – | 52.39 | 61.66 | – | – | 61.66 |
| Clipped | 47.90 | 49.20 | 50.90 | 49.33 | 55.57 | 54.60 | 58.38 | 56.18 | 58.90 | 60.80 | 62.20 | 60.63 |
| Standard | 43.57 | 46.50 | 46.50 | 45.52 | 50.64 | 53.22 | 53.68 | 52.51 | 52.96 | 60.43 | 62.64 | 58.68 |
| Auto-Regressive Large Language Models | | | | | | | | | | | | |
| Qwen2.5-7B-Instruct | 47.85 | 46.30 | 45.10 | 46.42 | 56.20 | 54.66 | 53.60 | 54.82 | 46.55 | 45.74 | 45.20 | 45.83 |
| Qwen3-8B-Base | **53.05** | **51.70** | **50.10** | **51.62** | **72.40** | **70.58** | **69.80** | **70.93** | 49.70 | 48.84 | 48.20 | 48.91 |
| Llama3.1-8B-Instruct | 44.95 | 47.07 | 47.76 | 46.59 | 47.60 | 46.40 | 49.10 | 47.70 | **65.02** | **66.28** | **64.88** | **65.39** |

Table 2: Per-seed performance of MMaDA-8B-MixCoT on text-to-image-2M. We report CLIP scores under three training seeds with identical model, data, and hyper-parameters. Avg is the mean CLIP over runs.

| Method (Seed) | text-to-image-2M | | | Avg |
|---|---|---|---|---|
| | 42 | 731 | 20231 | |
| P-POTS+Mirror | **35.27** | 34.10 | **34.39** | **34.59** |
| Standard | 28.61 | **34.28** | 31.19 | 31.36 |
| P-POTS | 31.51 | 30.02 | 33.41 | 31.65 |
| Mirror | 34.88 | 33.77 | 33.57 | 34.07 |

## 4.1 KEY INSIGHTS

MDMs exhibit uniquely high training variance, which makes gradient updates noisy and optimization less stable. As a result, even pretrained MDMs are shown to achieve comparable performance with strong ARMs, after task-specific training, they often under-perform by a wide margin. This instability is evident in Table 1: under standard MDM training on HiTab, accuracy varies substantially across random seeds (about $52\% \sim 63\%$), and the mean performance across datasets frequently remains below that of ARMs. Our method targets this root cause by reducing MDM training variance. Figure 4 shows that P-POTS+MIRROR (green) converges to a lower final loss with smoother dynamics than the Standard method (red), indicating more stable training. This improved stability shows consistent improvement and closes the training gap effectively: on all three textual benchmarks, even the best Standard/Clipped run remains below the worst run of our method (OpenScience: best Standard $46.50$ *vs.* worst P-POTS+MIRROR $50.80$; GSM8K: $53.68$ *vs.* $58.61$; HiTab: $62.64$ *vs.* $66.02$). On text-to-image-2M, the two methods are essentially comparable (best Standard $34.28$ *vs.* worst ours $34.10$). This highlights the importance of variance reduction in delivering more consistent and reliable performance across runs. Overall, MDMs trained with our method can match, and in some cases surpass, strong ARMs, while run-to-run variability drops to near ARM levels.

Moreover, ablation studies in A.8 confirm synergy between P-POTS and MIRROR. On GSM8K and OpenScience, compared to standard training, the accuracy gains from P-POTS+MIRROR ($8.02\%$ and $7.01\%$ respectively) surpass the sum of the individual gains from P-POTS and MIRROR (($6.07 + 1.19)\% = 7.26\%$ and $(1.28 + 0.86)\% = 2.14\%$ ), while on HiTab they are nearly additive ($8.42\%$

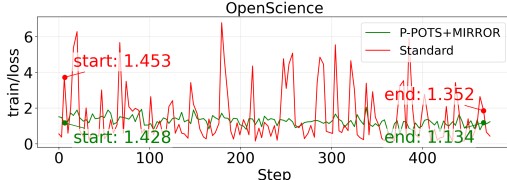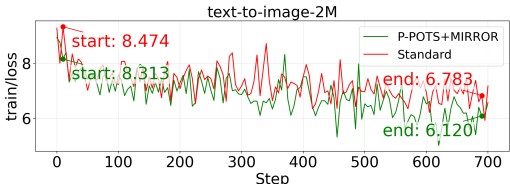

Figure 4: Training loss comparison between P-POTS+MIRROR (green) and Standard (red) on OpenScience (left) and text-to-image-2M (right). The annotations indicate the average loss at the beginning and end of training over the first and last 5 steps. Overall, P-POTS+MIRROR achieves more stable loss trajectories with consistently lower end losses compared to the Standard baseline.

and $(2.69 + 5.80)\% = 8.49\%$ ). We emphasize that this is nontrivial, because as discussed in A.6, naive combinations of methods do not typically yield such improvements.

MIRROR is especially effective on long-response datasets due to its design. As shown in Tables 1 and 2, its gains on OpenScience, HiTab and text-to-image-2M are substantially higher than on GSM8K. This aligns with its design: MIRROR targets masking pattern noise Ⓐ, which becomes more severe with longer responses due to the greater number of possible masking patterns.

MDMs achieve comparable performance as ARMs with much shorter outputs. In our experiments, MDMs often generate responses as short as 5% of ARM outputs and consistently shorter than training labels (see exact values in A.7.1). Truncating ARMs to the same length severely hurts their performance because outputs are often cut off mid-reasoning. One possible explanation is that MDMs decode by repeatedly unmasking a block of [MASK] tokens whose length is predetermined by the user-passed max_new_tokens, giving the model an implicit sense of how many tokens it is allowed to generate. In contrast, ARMs generate token by token without an explicit stopping criterion until hitting EOS, which can lead to verbosity or redundancy.

On text-to-image-2M, the drift in $p$ is more severe; see A.8.2. We attribute this to larger loss scales (and thus gradients) that move $\theta$ faster. Although EPR targets a U-shape, we use it as a stable compromise because P-POTS fits $p(t)$ only once pre-training; empirically, EPR is more robust than fitting the initial S-shape, so we recommend it as a default. In principle, periodically refitting $p(t)$ (*e.g.*, at each eval) could further improve P-POTS and P-POTS+MIRROR.

## 4.2 CASE STUDY

In Figure 3, standard training often produces blurry images or misses fine details. We suspect two causes. First, rare detail-critical tokens are masked too infrequently under uniform $p(t)$, so the model under-trains on details. Consequently, features such as the "crochet headband" and "toy sword" in case (b) are missing or distorted. Second, cross-entropy on masked image tokens can show an averaging effect. When multiple outputs are possible, the model may learn to predict their mean; at inference with low temperature, argmax decoding turns this into unstable choices, leading to blur. This may explain case (d), where fire and smoke looks unrealistic. By contrast, P-POTS+MIRROR may address these problems. P-POTS emphasizes harder regions, while MIRROR averages two negatively correlated samples and reduces the tendency to hedge. Together, they could improve the gradient signal-to-noise ratio, giving sharper textures and more faithful objects, *e.g.* correct headband/sword in (a) and more realistic flames in (c).

## 5 CONCLUSION

This work derives a systematic decomposition of MDM training variance as a fundamental framework for variance reduction, and proposes six corresponding techniques. Among them, P-POTS is provably Pareto-optimal for minimizing loss variance, while MIRROR is particularly beneficial on long-response datasets; together, they yield strong synergy. Our methods reduce both run-to-run variability and within-run suboptimality, making one training run sufficient for reliable performance. Looking forward, we expect our decomposition and methods to serve as building blocks for scalable MDM training, and to inspire future work toward a unified framework for variance control in masked diffusion models.

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

# A APPENDIX

CONTENTS

## A.1 STATEMENT

**Reproducibility.** All models and datasets used in this work are publicly available, as listed in 4. Details of dataset splits and hyperparameter configurations are provided in Appendix A.7.1, while the evaluation protocols for inference results are described in Appendix 4. All theoretical proofs and assumptions are presented in Appendix A.5.

**LLM Usgae.** We used large language models (LLMs) as a general-purpose assist tool in three ways: (i) polishing the writing for grammar and clarity, (ii) helping to identify potentially relevant

related work (with all citations manually verified by the authors), and (iii) providing code assistance for implementing ideas developed by the authors. The research ideas, methodology, experiments, and final paper content remain the responsibility of the authors.

## A.2 PRELIMINARIES

### A.2.1 STOCHASTIC GRADIENT DESCENT ON CONVEX OBJECTIVES

We consider the optimization problem

$$\min_{\beta \in C} f(\beta), \tag{6}$$

where $C \subseteq \mathbb{R}^p$ is a closed convex set and $f : C \to \mathbb{R}$ is convex. In machine learning, $f$ typically represents the population risk, *i.e.*, the expected loss with respect to the true but unknown data distribution. A common example is the expected cross-entropy loss of the model under parameter $\beta$ relative to the true data distribution.

Gradient-based optimization methods generally rely on estimates of the gradient of the expected loss. Using a larger batch size reduces the variance of this estimate but increases the computational cost per iteration. Stochastic Gradient Descent (SGD) strikes a balance by sampling a batch of fixed size at random and using the average loss gradient over this batch as an unbiased estimator of the population gradient. Convergence of SGD can be guaranteed in principle, provided that certain standard assumptions are satisfied:

**Theorem 1** Suppose $\hat{\beta}$ is a minimiser of $f$ over a closed convex set $C \subseteq \mathbb{R}^p$. Suppose $\sup_{\beta \in C} \|\beta\|_2 \leq R < \infty$ and

$$\sup_{\beta \in C} \mathbb{E} \left( \sup_{\tilde{g} \in \partial \tilde{f}(\beta; U)} \|\tilde{g}\|_2^2 \right) \leq L^2 < \infty.$$

Then if $\eta_s \equiv \eta = 2R/(L\sqrt{k})$, the output $\bar{\beta}$ of the stochastic gradient descent algorithm satisfies

$$\mathbb{E}f(\bar{\beta}) - f(\hat{\beta}) \ \leq \ \frac{2LR}{\sqrt{k}}.$$

**Note.** From a statistic point of view, the batch mean is merely an estimator of the population gradient, and thus inevitably subject to estimation variance. In MDM training, where the expected loss is subject to three sources of variance, this issue becomes particularly severe. It is therefore important to develop estimators that are more variance-efficient.

### A.2.2 DISCRETE DIFFUSION MODELS

Let $\mathbf{x}_0 \in \{1, \ldots, K\}^n$ denote an initial discrete data vector, such as a length $n$ token sequence, where each element $x_0^i$ takes values from a finite vocabulary of size $K$. A discrete diffusion model defines a forward corruption process that transforms $\mathbf{x}_0$ into a sequence of progressively noisier variables $\mathbf{x}_1, \ldots, \mathbf{x}_T$ over $T$ timesteps.

At each timestep $t = 1, \ldots, T$, let $x_t(k)$ represents the $k$th token of $\mathbf{x}_t$ for $k \in \{1, ..., n\}$, then each token $x_{t-1}(k)$ is corrupted into $x_t(k)$ using a predefined transition matrix $Q_t \in \mathbb{R}^{K \times K}$. Formally, the transition probabilities are defined as:

$$q(x_t(k) = j \mid x_{t-1}(k) = i) = [Q_t]_{i,j}. \tag{7}$$

where $Q_t$ is a stochastic matrix with each column sum to 1. Let $\mathbf{v}(x) \in R^K$ be a one-hot vector with 1 at position $x$. For simplicity, we abbreviate token position information and use $x_t$ for $\mathbf{x}_t(k)$ for any $k \in \{1, ..., K\}$. Then we can write the transition equation as

$$q(x_t \mid x_{t-1}) = \mathbf{v}^T(x_t) Q_t \mathbf{v}(x_{t-1}). \tag{8}$$

We can then marginalize intermediate steps and compute the transition probability from $x_0$ to $x_t$:

$$q(x_t \mid x_0) = \mathbf{v}^T(x_t)\tilde{Q}\mathbf{v}(x_0) = \mathbf{v}^T(x_t)Q_t \dots Q_1\mathbf{v}(x_0). \tag{9}$$

The posterior distribution conditioned on $x_0$ is then:

$$q(x_{t-1}|x_t, x_0) = \frac{q(x_t|x_{t-1}, x_0)q(x_{t-1}|x_0)}{q(x_t|x_0)} = \frac{(\mathbf{v}^T(x_t)Q_t\mathbf{v}(x_{t-1}))(\mathbf{v}^T(x_{t-1})\hat{Q}_{t-1}\mathbf{v}(x_0)))}{\mathbf{v}^T(x_t)\hat{Q}_t\mathbf{v}(x_0)} \tag{10}$$

For the reverse (denoising) process, a neural network $p_\theta(x_{t-1} \mid x_t) = \text{Categorical}(x_{t-1}; P_\theta(x_t))$ is trained to undo this corruption, where $P_\theta(x_t) \in \mathbb{R}^K$ denotes the logits produced by the model. $p_\theta(x_{t-1} \mid x_t)$ is trained to estimate this posterior. Since this posterior is defined on predefined forward processes only (the transition matrix $Q_T$ that controls the noisy level at each time step $t$), the design of transition matrix is crucial in the success of discrete diffusion models. A well-designed noise schedule ensures the model encounters a smooth curriculum of denoising tasks, leading to better final performance.

Early works propose a uniform noise design:

$$Q_t = \alpha_t I + (1 - \beta_t)\mathbf{1}\mathbf{1}^T, \beta_t = \frac{1 - \alpha_t}{K}$$

This design means at each time step, a token has probability $\alpha_t + \beta_t$ to remain and $\beta_t$ probability to transfer to other $K - 1$ tokens. This matrix has one eigenvalue $\lambda_1 = 1$(with eigenvector $\mathbf{1}$) and all other $K - 1$ eigenvalues equal to $\alpha_t$. Thus, the signal of $x_0$ decays like $\alpha_t...\alpha_0$. If this goes to 0 too quickly, you lose all information.

Among the general discrete diffusion model framework, masked diffusion models consistently achieves the best performance (Austin et al., 2021)(Lou et al., 2023). In masked diffusion, a specific design is employed in which the $K$th category serves as an absorbing state, commonly corresponding to a special token such as [MASK]. Under this configuration, the transition matrix $Q_t$ is defined as:

$$Q_t = \begin{bmatrix} \alpha_t & 0 & \cdots & 0 & 1 - \alpha_t \\ 0 & \alpha_t & \cdots & 0 & 1 - \alpha_t \\ \vdots & \vdots & \ddots & \vdots & 1 - \alpha_t \\ 0 & 0 & \cdots & \alpha_t & 1 - \alpha_t \end{bmatrix}.$$

where $\alpha_t \in [0, 1]$ denotes the masking ratio at timestep $t$. This design ensures that each token is independently replaced with the [MASK] token with probability $1-\alpha_t$, and retained with probability $\alpha_t$. Note the [MASK] token is an absorbing state. By isolating noise into a dedicated [MASK] token, the model preserves a clean, low-dimensional signal subspace, which it can focus on when reconstructing $\mathbf{x}_{t-1}$.

The training objective maximizes a variational lower bound (ELBO) on $\log p_\theta(x_0)$, as introduced in Eq. 1. Importantly, Sahoo et al. (2024) has shown that Eq. 1:

1. is invariant to the specific choice of the noise schedule $\alpha_t$, and
2. serves as an upper bound of

$$-\mathbb{E}_{p_{\text{data}}(x)}\big[p_\theta(x)\big],$$

thus enabling a principled and simplified training framework for masked diffusion models.

Restricting attention to *interpolating* masked diffusion models—*i.e.*, those whose forward process $q$ interpolates between clean data $x \in \mathcal{V}$ and a target distribution $\text{Cat}(\cdot; \pi)$, Sahoo et al. (2024) derived a principled training objective Eq. 1 that is invariant to noisy schedules. Recent models such as LLaDA and MMaDA base their training on this training objective and adopt a log-linear noise schedule defined by:

$$\alpha_t = \exp(\log(1 - t)) = 1 - t,$$

which means that at each timestep $t$, every token is independently replaced with a [MASK] token with probability $t$.

### A.2.3 PROXIMAL POLICY OPTIMIZATION

Proximal Policy Optimization (PPO) can be derived as a first-order approximation to the Trust Region Policy Optimization (TRPO) problem by replacing the hard KL-constraint with a KL-penalty. We begin with the constrained optimization

$$\max_{\theta} \quad \mathbb{E}_t\big[r_t(\theta)\,\hat{A}_t\big] \quad \text{s.t.} \quad \mathbb{E}_t\big[D_{\mathrm{KL}}\big[\pi_{\theta_{\mathrm{old}}}(\cdot \mid s_t) \,\|\, \pi_\theta(\cdot \mid s_t)\big]\big] \leq \delta, \tag{11}$$

where

$$r_t(\theta) \;=\; \frac{\pi_\theta(a_t \mid s_t)}{\pi_{\theta_{\mathrm{old}}}(a_t \mid s_t)} \quad \text{and} \quad \hat{A}_t \text{ is an estimator of the advantage at time } t.$$

Introducing a Lagrange multiplier $\beta$ for the KL-constraint yields the penalized surrogate objective

$$L^{\mathrm{KLPEN}}(\theta) \;=\; \mathbb{E}_t\Big[r_t(\theta)\,\hat{A}_t \;-\; \beta\, D_{\mathrm{KL}}\big[\pi_{\theta_{\mathrm{old}}}(\cdot \mid s_t) \,\|\, \pi_\theta(\cdot \mid s_t)\big]\Big]. \tag{12}$$

In practice, PPO proceeds by:

1. Collect trajectories $\{(s_t, a_t, r_t)\}$ under the old policy $\pi_{\theta_{\mathrm{old}}}$.

2. Compute advantage estimates $\hat{A}_t$ (*e.g.*, via GAE).

3. For $K$ epochs, perform minibatch gradient ascent on $L^{\mathrm{KLPEN}}(\theta)$:

$$\theta \;\leftarrow\; \theta \;+\; \eta\,\nabla_\theta L^{\mathrm{KLPEN}}(\theta).$$

4. Optionally adjust $\beta$ to keep the KL close to a target $\delta$:

$$\beta \;\leftarrow\; \begin{cases} \alpha_+\,\beta, & \bar{D}_{D_{\mathrm{KL}}} > 1.5\,\delta, \\ \alpha_-\,\beta, & \bar{D}_{D_{\mathrm{KL}}} < \delta/1.5, \\ \beta, & \text{otherwise}, \end{cases}$$

where $\bar{D}_{D_{\mathrm{KL}}} = \mathbb{E}_t[D_{\mathrm{KL}}(\pi_{\theta_{\mathrm{old}}}\|\pi_\theta)]$.

5. Set $\theta_{\mathrm{old}} \leftarrow \theta$ and repeat.

### A.2.4 GROUP RELATIVE POLICY OPTIMIZATION

Group Relative Policy Optimization (GRPO) extends PPO to settings where we have a collection of $G$ groups (or tasks), each potentially requiring its own policy behavior while sharing parameters. Let $\mathcal{G} = \{1, \ldots, G\}$ denote the set of groups, with weightings $\{\omega_g\}_{g \in \mathcal{G}}$ such that $\sum_g \omega_g = 1$. For each group $g$, define

$$r_t^g(\theta) \;=\; \frac{\pi_\theta(a_t \mid s_t, g)}{\pi_{\theta_{\mathrm{old}}}(a_t \mid s_t, g)}, \quad \hat{A}_t^g \;=\; \text{advantage estimate for group } g \text{ at time } t,$$

and the group-specific KL divergence

$$D_{D_{\mathrm{KL}}}^g(\theta) = \mathbb{E}_{t \sim g}\Big[D_{\mathrm{KL}}\big[\pi_{\theta_{\mathrm{old}}}(\cdot \mid s_t, g) \,\|\, \pi_\theta(\cdot \mid s_t, g)\big]\Big].$$

Introducing a common penalty coefficient $\beta$, the GRPO surrogate objective is

$$L^{\mathrm{GRPO}}(\theta) \;=\; \sum_{g \in \mathcal{G}} \omega_g\, \mathbb{E}_{t \sim g}\Big[r_t^g(\theta)\,\hat{A}_t^g \;-\; \beta\, D_{\mathrm{KL}}\big[\pi_{\theta_{\mathrm{old}}}(\cdot \mid s_t, g) \,\|\, \pi_\theta(\cdot \mid s_t, g)\big]\Big]. \tag{13}$$

In some cases, one may also include an inter-group regularizer to encourage policies for different groups to remain similar:

$$-\,\gamma \sum_{\substack{g, h \in \mathcal{G} \\ g \neq h}} \omega_{g,h}\, \mathbb{E}_{s \sim \mathcal{D}}\Big[D_{\mathrm{KL}}\big[\pi_\theta(\cdot \mid s, g) \,\|\, \pi_\theta(\cdot \mid s, h)\big]\Big],$$

where $\gamma \geq 0$ and $\{\omega_{g,h}\}$ are pairwise weights.

The GRPO algorithm proceeds analogously to PPO:

1. For each $g \in \mathcal{G}$, collect trajectories under $\pi_{\theta_{\text{old}}}(\,\cdot\mid\cdot,g)$.

2. Compute group-specific advantages $\{\hat{A}_t^g\}$.

3. For $K$ epochs, perform minibatch gradient ascent on $L^{\text{GRPO}}(\theta)$:

$$\theta \;\leftarrow\; \theta \;+\; \eta\,\nabla_\theta L^{\text{GRPO}}(\theta).$$

4. Optionally adjust $\beta$ (and $\gamma$, if used) to keep each $D_{D_{\text{KL}}}^g$ (and inter-group KLs) near desired targets.

5. Update $\theta_{\text{old}} \leftarrow \theta$ and repeat.

### A.3 TRAINING VARIANCE DECOMPOSITION

The proof of Eq. 2 is deferred to A.5.1. Here we clarify why this decomposition is well-motivated and how it supports the P-POTS derivation (§3.1.1).

**Effect of Alternative decompositions.** Our decomposition is obtained by iteratively conditioning on the random variables $(x_0, t, x_t)$, so different conditioning orders give different decompositions. The order adopted in this paper $x_0 \to t \to x_y$ is the only interpretable one, because it aligns with the MDM training procedure and matches ARMs' variance decomposition. Importantly, the resulting form of P-POTS in § 3.1.1 is invariant to the conditioning order. This is because the optimal sampler $p^*(t)$ depends only on $\mathbb{E}[l^2 \mid t]$, not on how the total variance is partitioned.

**Loss Variance *vs.* Gradient Variance.** While gradient variance is most directly tied to optimization stability, we focus on loss variance for three reasons. (1) it is is optimizer-agnostic: adaptive optimizers rescale gradients, making "raw" gradient variance dependent on the optimizer, whereas $\text{Var}(l)$ directly improves the SNR of the Monte Carlo estimate of the training objective. (2) Gradients are high-dimensional (*e.g.*, a 8B-model has a gradient of dimension 8B), so any notion of "gradient variance" requires a scalarization (*e.g.*, $\mathbb{E}\|\mathbf{g}\|_2^2$). Yet this is not theoretically neutral: different parameters may prefer different sampling distributions. Any scalarization therefore introduce extra design choices and breaks the clean optimality claim. In contrast, the loss is already a unified scalar with a clear interpretation as the objective whose expectation we estimate, so its variance is unambiguous. (3) A $t$-sampler can reliably control only the conditional variance of the loss: by first-order linearization, $\text{Var}(\mathbf{g} \mid t) \approx J_t\,\text{Var}(l \mid t)\,J_t^\top + $ (terms from model sensitivity), where $J_t$ is the Jacobian *w.r.t.* parameters. This highlights that part of gradient variance comes from model sensitivity (the Jacobian), which cannot be affected by $t$-sampler. The only reliably controllable component is $\text{Var}(l \mid t)$ (holding fixed over short horizons).

### A.4 ALGORITHMS

#### A.4.1 BIN-WISE EMA

We partition the interval $t \in [0, 1]$ into $m$ equal bins and fix an EMA learning rate $\eta$. For each bin $j$, we maintain four EMA-tracked quantities:

$$\mu_L(j) \approx \mathbb{E}[\ell_i \mid t \in \text{bin } j], \quad \mu_H(j) \approx \mathbb{E}[\tilde{b}_j], \quad M_{LH}(j) \approx \mathbb{E}[\ell_i \tilde{b}_j], \quad M_{HH}(j) \approx \mathbb{E}[\tilde{b}_j^2],$$

and a per-bin baseline estimate $\tilde{b}_j$, initialized to zero. On each training step:

- Draw a batch $\{x_i\}_{i=1}^B$, sample mask ratios $t_i \sim \mathcal{U}[0, 1]$, run the forward diffusion, and compute per-sample losses $\ell_i$.

- Assign each $t_i$ to bin $j$ via

$$j = f(t_i) \quad \text{if } t_i \in \left(\frac{j-1}{m}, \frac{j}{m}\right].$$

- For each sample, form an online estimate of the optimal control-variate coefficient

$$c_j = \frac{M_{LH}(j) - \mu_L(j)\mu_H(j)}{M_{HH}(j) - \mu_H(j)^2 + \epsilon},$$

where $\epsilon$ is a small damping constant.

- Compute the adjusted loss

$$\ell_i^{\text{adj}} = \ell_i - c_j \cdot \tilde{b}_j,$$

  detaching $\tilde{b}_j$ so that gradients flow only through $\ell_i$, and backpropagate on $\ell_i^{\text{adj}}$.

- Update all four EMA statistics for bin $j$:

$$\mu_L(j) \leftarrow (1-\eta)\mu_L(j) + \eta\ell_i,$$
$$\mu_H(j) \leftarrow (1-\eta)\mu_H(j) + \eta\tilde{b}_j,$$
$$M_{LH}(j) \leftarrow (1-\eta)M_{LH}(j) + \eta(\ell_i\tilde{b}_j),$$
$$M_{HH}(j) \leftarrow (1-\eta)M_{HH}(j) + \eta(\tilde{b}_j^2).$$

- Finally, update the baseline itself via

$$\tilde{b}_j \leftarrow (1-\eta)\tilde{b}_j + \eta\ell_i.$$

This procedure produces a piecewise-constant approximation $\tilde{\mathbf{b}}$ of the true expected loss $b(t)$ while dynamically choosing, per bin, the coefficient $c_j$ that maximally cancels out baseline-related noise–yielding tighter variance reduction than a fixed scaling *e.g.*, $c = 1$. Additionally, this "per-bin" EMA is updated per GPU batch and is analogous to standard moving-average updates used elsewhere in deep learning.

### A.5 PROOFS

#### A.5.1 TRAINING VARIANCE DECOMPOSITION

The law of total variance states that

$$\text{Var}(Y) = \mathbb{E}\big[\text{Var}(Y \mid X)\big] + \text{Var}\big(\mathbb{E}[Y \mid X]\big). \tag{14}$$

Let $Y = l(x_0, t, x_t)$ where $x_0 \sim p_{\text{data}}(.), t \sim U[0,1], x_t \sim q(x_t \mid x_0, t)$ are three random variables. We apply the law of total variance twice:

**1. First Decomposition over $x_0$**

View $x_0$ as the outer random variable. By the law of total variance,

$$\text{Var}_{x_0,t,x_t}(Y) = \mathbb{E}_{x_0}\left[\text{Var}_{t,x_t}(Y \mid x_0)\right] + \text{Var}_{x_0}\left[\mathbb{E}_{t,x_t}(Y \mid x_0)\right].$$

Define

$$g_\theta(x_0, t) = \mathbb{E}_{x_t}[\ell_\theta(x_0, t, x_t) \mid x_0, t], \quad h_\theta(x_0) = \mathbb{E}_{t,x_t}[\ell_\theta(x_0, t, x_t) \mid x_0] = \mathbb{E}_t[g_\theta(x_0, t) \mid x_0].$$

Then the second term becomes

$$\text{Var}_{x_0}[h_\theta(x_0)] = \text{Var}_{x_0}\left(\mathbb{E}_t[g_\theta(x_0, t)]\right) \qquad \text{Ⓒ data variance.}$$

**2. Second decomposition inside $\mathbb{E}_{x_0}[\text{Var}_{t,x_t}(Y \mid x_0)]$**

We apply the law of total variance again, this time conditioning on $t$:

$$\text{Var}_{t,x_t}(Y \mid x_0) = \mathbb{E}_t\left[\text{Var}_{x_t}(Y \mid x_0, t)\right] + \text{Var}_t\left[\mathbb{E}_{x_t}(Y \mid x_0, t)\right].$$

Note that $\mathbb{E}_{x_t}(Y \mid x_0, t) = g_\theta(x_0, t)$, so:

$$\mathbb{E}_{x_0}[\text{Var}_{t,x_t}(Y \mid x_0)] = \mathbb{E}_{x_0}\left[\mathbb{E}_t\left[\text{Var}_{x_t}(\ell_\theta \mid x_0, t)\right]\right] + \mathbb{E}_{x_0}\left[\text{Var}_t(g_\theta(x_0, t) \mid x_0)\right],$$
$$= \underbrace{\mathbb{E}_{x_0,t}\left[\text{Var}_{x_t}(\ell_\theta \mid x_0, t)\right]}_{\text{Ⓐ mask pattern noise}} + \underbrace{\mathbb{E}_{x_0}\left[\text{Var}_t(g_\theta(x_0, t) \mid x_0)\right]}_{\text{Ⓑ mask rate noise}}.$$

**3. Combine Everything**

Putting steps 1 and 2 together, we have:

$$\text{Var}_{x_0,t,x_t}(\ell_\theta) = \underbrace{\mathbb{E}_{x_0,t}\left[\text{Var}_{x_t}(\ell_\theta \mid x_0, t)\right]}_{\text{Ⓐ}} + \underbrace{\mathbb{E}_{x_0}\left[\text{Var}_t(g_\theta(x_0, t) \mid x_0)\right]}_{\text{Ⓑ}} + \underbrace{\text{Var}_{x_0}\left(\mathbb{E}_t[g_\theta(x_0, t)]\right)}_{\text{Ⓒ}},$$

which is exactly the claimed decomposition Eq. 2.

### A.5.2   P-POTS: PARETO-OPTIMAL PROPERTY

Consider the reweighted estimator

$$\frac{1}{p(t)} \, l_\theta(x_0, t, x_t), \quad t \sim p(t).$$

As shown in Eq. 3, this estimator is unbiased for the training objective in Eq. 1. We now give a detailed proof that

$$p(t) = \frac{\sqrt{g^2(t) + v(t)}}{\int_0^1 \sqrt{g^2(s) + v(s)} \, ds}$$

is the theoretically optimal distribution that minimizes $\textcircled{A} + \textcircled{B} + \textcircled{C}$.

Recall $\mathrm{Var}[X] = \mathbb{E}[X^2] - \mathbb{E}[X]^2$. Then

$$\textcircled{A} + \textcircled{B} + \textcircled{C} = \mathrm{Var}_{x_0, t, x_t}\Big(\tfrac{1}{p(t)} \, l_\theta(x_0, t, x_t)\Big).$$

Expanding yields

$$\mathbb{E}_{x_0, t, x_t}\Big[\tfrac{1}{p^2(t)} \, l_\theta^2(x_0, t, x_t)\Big] - \mathbb{E}_{x_0, t, x_t}^2\Big[\tfrac{1}{p(t)} \, l_\theta(x_0, t, x_t)\Big].$$

Assuming absolute convergence and finite expectations, Fubini's theorem allows us to exchange the order of integration, giving

$$\int_0^1 \frac{1}{p(t)} \int_{x_0 \sim D} \int_{X_t} l_\theta^2(x_0, t, x_t) \, dx_t dx_0 \, dt - \left(\int_0^1 \int_{x_0 \sim D} \int_{X_t} l_\theta(x_0, t, x_t) \, dx_t dx_0 \, dt\right)^2.$$

Define

$$g(t) := \mathbb{E}_{x_0, x_t}[l_\theta \mid t], \qquad v(t) := \mathrm{Var}_{x_0, x_t}[l_\theta \mid t].$$

Substituting yields

$$\int_0^1 \frac{g^2(t) + v(t)}{p(t)} \, dt - \left(\int_0^1 g(t) \, dt\right)^2.$$

The second term is independent of $p(t)$. Thus the optimization problem reduces to

$$\min_p \; J[p] := \int_0^1 \frac{g^2(t) + v(t)}{p(t)} \, dt, \quad \text{s.t.} \; p(t) \geq 0, \; \int_0^1 p(t) \, dt = 1.$$

**Lagrangian method.**   Form the Lagrangian:

$$\mathcal{L}[p, \lambda] = \int_0^1 \Big(\frac{g^2(t) + v(t)}{p(t)} + \lambda p(t)\Big) \, dt - \lambda,$$

where $\lambda$ enforces normalization. Taking the variation *w.r.t.* $p(t)$ gives

$$-\frac{g^2(t) + v(t)}{p(t)^2} + \lambda = 0 \quad \Longrightarrow \quad p^*(t) = \sqrt{\tfrac{g^2(t) + v(t)}{\lambda}}.$$

Normalization requires

$$\sqrt{\lambda} = \int_0^1 \sqrt{g^2(s) + v(s)} \, ds.$$

**Final result.**   Therefore, the unique minimizer is

$$\boxed{p^*(t) = \frac{\sqrt{g^2(t) + v(t)}}{\int_0^1 \sqrt{g^2(s) + v(s)} \, ds}}$$

which achieves the Pareto-optimal balance by minimizing $\textcircled{A} + \textcircled{B} + \textcircled{C}$.

### A.5.3 IMPORTANCE SAMPLING

**Part 1: Unbiasedness of the Importance-Sampling Estimator**

Suppose we wish to estimate

$$I = \int f(x)\, p(x)\, dx,$$

but drawing samples from $p(x)$ directly is difficult. Instead, we draw $x_1, \ldots, x_N$ i.i.d. from a proposal distribution $q(x)$, which satisfies $q(x) > 0$ whenever $f(x)p(x) \neq 0$, and construct the estimator

$$\hat{I}_N = \frac{1}{N} \sum_{i=1}^{N} w(x_i)\, f(x_i), \quad \text{where } w(x) = \frac{p(x)}{q(x)}.$$

We check its expectation under the sampling law $q$:

$$
\begin{aligned}
\mathbb{E}_q[\hat{I}_N] &= \frac{1}{N} \sum_{i=1}^{N} \mathbb{E}_q[w(x_i)f(x_i)] \quad \text{(by linearity and identical distribution)} \\
&= \mathbb{E}_q[w(X)f(X)] \quad \text{(drop index)} \\
&= \int \frac{p(x)}{q(x)} f(x)q(x)\, dx = \int f(x)p(x)\, dx = I.
\end{aligned}
$$

Hence, $\hat{I}_N$ is an unbiased estimator of $I$.

**Part 2: Choice of $q$ to Minimize Variance**

The variance of $\hat{I}_N$ under $q$ is

$$\mathrm{Var}_q(\hat{I}_N) = \frac{1}{N}\, \mathrm{Var}_q[w(X)f(X)] = \frac{1}{N}\left(\mathbb{E}_q[w(X)^2 f(X)^2] - I^2\right).$$

Since $I$ is fixed, minimizing $\mathrm{Var}_q(\hat{I}_N)$ amounts to minimizing

$$\mathbb{E}_q[w(X)^2 f(X)^2] = \int \frac{p(x)^2}{q(x)} f(x)^2\, dx,$$

subject to the constraint $\int q(x)\, dx = 1$.

Introduce a Lagrange multiplier $\lambda$, and consider the functional

$$\mathcal{L}[q] = \int \frac{p(x)^2 f(x)^2}{q(x)}\, dx + \lambda \left(\int q(x)\, dx - 1\right).$$

Taking the functional derivative with respect to $q(x)$ and setting it to zero:

$$-\frac{p(x)^2 f(x)^2}{q(x)^2} + \lambda = 0 \quad \Rightarrow \quad q(x) \propto p(x)\,|f(x)|.$$

Thus, the optimal proposal distribution is

$$q^*(x) = \frac{|f(x)|\, p(x)}{\int |f(u)|\, p(u)\, du},$$

which, for nonnegative $f$, minimizes the variance to its theoretical minimum–zero in the ideal continuous-sampling limit.

### A.5.4   SYRM

SyRM is designed for structured data such as HTML tables, source code, or graphs. We categorize tokens into two groups: 1) response tokens $R$, and 2) syntax tokens $C$ in the prompt. For example, in an html table, the syntax tokens are `<table>`, `<tr>`, `<th>`, `<td>`, and similar tags. SyRM modifies the eligibility set for masking as follows:

$$\text{eligible} := \text{response tokens} \cup \text{ syntax tokens in prompt}$$

**Definition of Eligible Tokens**

We define a token position as *eligible* if it can be masked during the forward process. According to LLaDA(Nie et al., 2025), during pretraining, all tokens in $\mathbf{x}_0$ are eligible. During supervised fine-tuning, where $\mathbf{x}_0$ is composed of concatenated prompt and response tokens, only response tokens are eligible. Thus, in this stage, prompt tokens remain unmasked while response tokens are masked independently.

**Under the following assumptions,**

- Tokens within each group are homogeneous in first and second moments of token-level losses.
- Syntax tokens are stabler predicted than response tokens.
- Reasoning errors couple response tokens more strongly than syntax tokens.

we show below that SyRM rigorously reduces the masking pattern noise Ⓐ. Although this introduces some bias (optimum shift), we show this bias is bounded, and the total MSE of estimating training loss is reduced due to significant variance reduction. Intuitively, by including syntax tokens in prompts that are both easy to predict and helpful for guiding the model's reasoning as an auxiliary loss, SyRM reduces Ⓐ mask pattern noise.

**Setup**

- Number of *eligible* positions: $P \in \mathbb{N}$.
- Mask variables: $M_i \overset{\text{i.i.d.}}{\sim} \text{Bernoulli}(t)$ with $0 < t < 1$.
- Per-token loss: random variables $Y_i$ satisfying

$$\mathbb{E}[Y_i] = \mu_i, \qquad \text{Var}(Y_i) = \sigma_i^2, \qquad \text{Cov}(Y_i, Y_j) = \rho_{ij} \ (i \neq j).$$

- Batch loss:

$$L = \frac{1}{Pt} \sum_{i=1}^{P} M_i Y_i.$$

- The $M_i$'s are independent of all $Y_j$, and the $M_i$'s are mutually independent.

**Theorem 1**

$$\text{Var}\big[L \mid t\big] = \frac{1}{P^2 t} \sum_{i=1}^{P} \sigma_i^2 + \frac{1-t}{P^2 t} \sum_{i=1}^{P} \mu_i^2 + \frac{2}{P^2} \sum_{1 \leq i < j \leq P} \rho_{ij}.$$

**Proof**   Let $S := \sum_{i=1}^{P} M_i Y_i$, so $L = S/(Pt)$.

Then

$$\text{Var}(S) = \sum_{i=1}^{P} \text{Var}(M_i Y_i) + 2 \sum_{1 \leq i < j \leq P} \text{Cov}(M_i Y_i, M_j Y_j). \tag{15}$$

where

$$\text{Var}(M_i Y_i) = \mathbb{E}[M_i Y_i^2] - (\mathbb{E}[M_i Y_i])^2 = t\mathbb{E}[Y_i^2] - t^2 \mu_i^2 = t\sigma_i^2 + t(1-t)\mu_i^2. \tag{16}$$

and

$$\mathrm{Cov}(M_iY_i, M_jY_j) = \mathbb{E}[M_iM_jY_iY_j] - \mathbb{E}[M_iY_i]\mathbb{E}[M_jY_j] = t^2(\rho_{ij} + \mu_i\mu_j) - t^2\mu_i\mu_j = t^2\rho_{ij}. \quad (17)$$

Combine Eqs. 15, 16 and 17:

$$\mathrm{Var}(S) = t\sum_{i=1}^{P}\sigma_i^2 + t(1-t)\sum_{i=1}^{P}\mu_i^2 + 2t^2\sum_{1\le i<j\le P}\rho_{ij}. \quad (18)$$

Convert Eq. 18 back to $L = \frac{S}{Pt}$:

$$\mathrm{Var}[L \mid t] = \frac{1}{(Pt)^2}\,\mathrm{Var}(S)$$

$$= \frac{1}{P^2t}\sum_{i=1}^{P}\sigma_i^2 + \frac{1-t}{P^2t}\sum_{i=1}^{P}\mu_i^2 + \frac{2}{P^2}\sum_{1\le i<j\le P}\rho_{ij}.$$

**Theorem 2** Let $S_{\mathrm{C}}$ be the SyRM strategy and $S_{\mathrm{R}}$ the baseline strategy that masks only response tokens. Define the mask-pattern noise of a strategy $S$ as $A(S) := \mathrm{Var}[L \mid S]$. For $m \in \{\mathrm{C}, \mathrm{R}\}$ with corresponding eligible-set size $P_m$ and token statistics $\mu_{i,m}, \sigma_{i,m}^2, \rho_{ij,m}$,

$$A(S_m) = \frac{\mathbb{E}[1/t]}{P_m^2}\sum_{i=1}^{P_m}\sigma_{i,m}^2 + \frac{\mathbb{E}[(1-t)/t]}{P_m^2}\sum_{i=1}^{P_m}\mu_{i,m}^2 + \frac{2}{P_m^2}\sum_{1\le i<j\le P_m}\rho_{ij,m}.$$

**Proof** Take the conditional variance from Theorem 1 and integrate over $t$. Note that token-wise parameters naturally depend on the chosen strategy, so we let $P, \sigma_i, \mu_i, \rho_{ij}$ depend on $m \in \{\mathrm{C}, \mathrm{R}\}$.

**Theorem 3** Partition the eligible tokens into

1. response tokens $R$ (count $P_R$),
2. syntax prompt tokens $C$ (count $P_C$),

so $P = P_R + P_C$. Under the assumptions

(A1) All tokens in $R$ share $\mu_R, \sigma_R^2$; tokens in $C$ share $\mu_C, \sigma_C^2$.

(A2) $\sigma_C^2 \ll \sigma_R^2$.

(A3) $\rho_{CC} \ll \rho_{RR}$.

(A4) $|\rho_{RC}| \le \sqrt{\rho_{RR}\rho_{CC}}$.

(A5) $2w_R\beta \le \dfrac{(1-\alpha)(P_R-1)\rho_{RR}}{\sigma_R^2 B} + (1-\beta)$, where $\alpha := \sigma_C^2/\sigma_R^2$, $\beta := \rho_{CC}/\rho_{RR}$, $w_R := P_R/P$, and $B := \mathbb{E}[1/t] > 0$,

we have

$$A(S_{\mathrm{C}}) < A(S_{\mathrm{R}}).$$

**Proof** Denote $B := \mathbb{E}\!\left[\frac{1}{t}\right] > 0$. By Theorem 2,

$$A(S_{\mathrm{R}}) = \frac{B}{P_R}\sigma_R^2 + \frac{1}{P_R}(B-1)\mu_m^2 + \frac{P_R-1}{P_R}\rho_{RR} \quad (19)$$

In the SyRM strategy, both R and C are *eligible*. Denote $w_R = \frac{P_R}{P}, w_C = \frac{P_C}{P}$ where $P = P_R + P_C > \max(P_R, P_C)$. We group terms accordingly, yielding

$$A(S_C) = \frac{\sigma_R^2}{P_R B} + \frac{\sigma_C^2}{P_C B} + \frac{P_R - 1}{P}\rho_{RR} + \frac{P_C - 1}{P}\rho_{CC} + \frac{2P_R P_C}{P^2}\rho_{RC} = w_R A_R + w_C A_C + 2w_R w_C \rho_{RC}, \tag{20}$$

where

$$A_R := \frac{\sigma_R^2}{P_R B} + \frac{P_R - 1}{P_R}\rho_{RR}, \qquad A_C := \frac{\sigma_C^2}{P_C B} + \frac{P_C - 1}{P_C}\rho_{CC}.$$

Thus,

$$A(S_C) - A(S_R) = w_C(A_C - A_R) + 2w_R w_C \rho_{RC}.$$

**Claim 3.1**

$$A_R > A_C$$

**Proof**  By the Cauchy–Schwarz inequality,

$$|\rho_{RC}| \le \sqrt{\rho_{RR}\rho_{CC}}.$$

From Eqs. 19 and 20, we further obtain:

$$A_R - A_C = \left(\frac{\sigma_R^2}{P_R} - \frac{\sigma_C^2}{P_C}\right)B + \left(\frac{P_R - 1}{P_R}\rho_{RR} - \frac{P_C - 1}{P_C}\rho_{CC}\right). \tag{21}$$

Define:

$$\alpha := \frac{\sigma_C^2}{\sigma_R^2}, \qquad \beta := \frac{\rho_{CC}}{\rho_{RR}} \quad \text{(as in Assumption A, assume } \alpha, \beta \ll 1\text{),}$$

then:

$$A_R - A_C = \sigma_R^2 B\left(\frac{1}{P_R} - \frac{\alpha}{P_C}\right) + \rho_{RR}\left(\frac{P_R - 1}{P_R} - \beta \cdot \frac{P_C - 1}{P_C}\right) \ge \sigma_R^2 B \cdot \frac{1 - \alpha}{P_R} + \rho_{RR} \cdot \frac{1 - \beta}{P_R} \cdot \frac{P_R - 1}{1}. \tag{22}$$

Therefore,

$$A_R > A_C.$$

Using the inequality

$$|\rho_{RC}| \le \sqrt{\rho_{RR}\rho_{CC}} \le \beta\rho_{RR},$$

we obtain:

$$A(S_C) - A(S_R) \le w_C \underbrace{\left[-(1 - \alpha)\frac{\sigma_R^2 B}{P_R} - (1 - \beta)\frac{P_R - 1}{P_R}\rho_{RR}\right]}_{=:D} + 2w_R w_C \beta\rho_{RR}.$$

Since $D > 0$ and $\beta \ll 1$, setting

$$2w_R\beta \le \frac{D}{\rho_{RR}},$$

ensures that

$$A(S_C) - A(S_R) < 0.$$

**Discussion of Assumptions**

- A1–A2. Syntax tokens $C$ are homogeneous and more stably predicted, hence lower mean and variance.

- A3. Reasoning errors couple response tokens more strongly than syntax tokens, implying $\rho_{RR} \gg \rho_{CC}$.

- A4. Cauchy–Schwarz bounds the cross-covariance.

- A5. Given A2–A3, this technical condition is typically satisfied in practice and ensures the cross-term cannot offset the variance reduction provided by including $C$.

**Theorem 4**

$$L_{\text{SyRM}} = \alpha\, L_{\text{resp}} + (1-\alpha)\, L_{\text{coord}}$$

**Proof 4**   Consider a single forward pass with $P_R$ response tokens and $P_C$ syntax tokens (disjoint sets, so $P = P_R + P_C$). All eligible positions are masked with the same rate $t \sim \text{Uniform}(0,1)$. Define the per-step loss

$$L \;=\; \frac{1}{Pt} \sum_{i=1}^{P} M_i Y_i, \qquad M_i \sim \text{Bernoulli}(t),$$

where $P$ is the number of eligible tokens under the chosen strategy.

For the SyRM strategy, $P_{\text{SyRM}} = P_R + P_C$, and

$$
\begin{aligned}
L_{\text{SyRM}} &= \frac{1}{P_{\text{SyRM}}\, t} \sum_{i \in R \cup C} M_i Y_i \\
&= \frac{P_R}{P_{\text{SyRM}}} \Big[ \frac{1}{P_R\, t} \sum_{i \in R} M_i Y_i \Big] + \frac{P_C}{P_{\text{SyRM}}} \Big[ \frac{1}{P_C\, t} \sum_{j \in C} M_j Y_j \Big] \\
&:= \alpha\, L_{\text{resp}} + (1-\alpha)\, L_{\text{coord}},
\end{aligned}
$$

where

$$\alpha := \frac{P_R}{P_R + P_C}, \qquad 1 - \alpha := \frac{P_C}{P_R + P_C}.$$

Hence,

$$L_{\text{SyRM}} \;=\; \alpha\, L_{\text{resp}} + (1-\alpha)\, L_{\text{coord}},$$

which is a purely algebraic identity requiring no additional assumptions.

**Theorem 5**   Optimum-shift bias is inevitable.

**Proof**   Define the objectives $J_{\text{resp}}(\theta) = \mathbb{E}[L_{\text{resp}}]$, $J_{\text{coord}}(\theta) = \mathbb{E}[L_{\text{coord}}]$, $J_{\text{SyRM}}(\theta) = \alpha\, J_{\text{resp}}(\theta) + (1-\alpha)\, J_{\text{coord}}(\theta)$. Let the response-only optimum $\theta^\star_{\text{resp}}$ satisfy $\nabla_\theta J_{\text{resp}}(\theta^\star_{\text{resp}}) = 0$. If at the same point $\nabla_\theta J_{\text{coord}}(\theta^\star_{\text{resp}}) \neq 0$, then

$$\nabla_\theta J_{\text{SyRM}}(\theta^\star_{\text{resp}}) = (1-\alpha)\, \nabla_\theta J_{\text{coord}}(\theta^\star_{\text{resp}}) \neq 0,$$

so $\theta^\star_{\text{resp}}$ is *not* a stationary point of $J_{\text{SyRM}}$. Under gradient-based optimization, the parameters will continue to move and eventually converge to a different optimum $\theta^\star_{\text{SyRM}} \neq \theta^\star_{\text{resp}}$. Hence, whenever $\nabla_\theta J_{\text{coord}}(\theta^\star_{\text{resp}}) \neq 0$, the two optima necessarily diverge; coincidence occurs iff $\nabla_\theta J_{\text{coord}}(\theta^\star_{\text{resp}}) = 0$.

**Theorem 6**   If the Hessian of $J_{\text{SyRM}}$ is positive definite,

$$\|\theta^\star_{\text{SyRM}} - \theta^\star_{\text{resp}}\| \leq \frac{1-\alpha}{\lambda_{\min}(H_{\text{SyRM}})} \, \big\|\nabla J_{\text{coord}}(\theta^\star_{\text{resp}})\big\|.$$

**Proof**   Perform a first-order Taylor expansion of $\theta_{\text{SyRM}}^\star$ around $\theta_{\text{resp}}^\star$:

$$0 = \nabla J_{\text{SyRM}}(\theta_{\text{SyRM}}^\star) \simeq \nabla J_{\text{SyRM}}(\theta_{\text{resp}}^\star) + H_{\text{SyRM}}(\xi)(\theta_{\text{SyRM}}^\star - \theta_{\text{resp}}^\star),$$

where $H_{\text{SyRM}}(\xi)$ is the Hessian evaluated at some point $\xi$ between $\theta_{\text{SyRM}}^\star$ and $\theta_{\text{resp}}^\star$. Rearranging gives:

$$\theta_{\text{SyRM}}^\star - \theta_{\text{resp}}^\star \simeq -(1-\alpha)H_{\text{SyRM}}^{-1}(\xi)\nabla J_{\text{coord}}(\theta_{\text{resp}}^\star).$$

If $H_{\text{SyRM}}(\xi)$ is positive definite, we can bound the Euclidean norm using its smallest eigenvalue $\lambda_{\min}(H_{\text{SyRM}})$:

$$\left\|\theta_{\text{SyRM}}^\star - \theta_{\text{resp}}^\star\right\| \leq \frac{1-\alpha}{\lambda_{\min}(H_{\text{SyRM}})}\left\|\nabla J_{\text{coord}}(\theta_{\text{resp}}^\star)\right\|.$$

**Explanation:**

- Weight factor $1-\alpha$: The smaller the proportion of syntax tokens, the tighter the bound.

- Curvature $\lambda_{\min}(H_{\text{SyRM}})$: A steeper loss (larger curvature) suppresses the shift.

- Task conflict $\left\|\nabla J_{\text{coord}}(\theta_{\text{resp}}^\star)\right\|$: If the syntax gradient is already small at the resp-optimal point, the shift will naturally be small.

### A.5.5   STRATIFIED SAMPLER

Specifically, when processing a batch of $n$ data points, instead of independently drawing mask ratios $t_i \sim \mathcal{U}[0,1]$, one partitions the interval $[0,1]$ into $k$ uniform strata: $[0,1/k], [1/k, 2/k], \ldots, [(k-1)/k, 1]$, and samples one masking ratio from each stratum. This approach, known as the *stratified sampler*, is known to produce low discrepancy. Here, $k$ is the number of strata, which must be chosen before training.

This technique is discussed in Sahoo et al. (2024), where they use $n = k$ (*i.e.*, one sample per stratum). Here, we provide a rigorous analysis and offer a practical guideline for selecting $k$.

**A principled Method: Unbiased and Reduce Variance**

Let

$$G(t) = g_\theta(x_0, t) = \mathbb{E}_{x_t \sim q(\cdot|x_0, t)}[\ell_\theta(x_0, t, x_t) \mid x_0, t].$$

For a fixed $x_0$, we are interested in the expected value of $G(t)$ under $t \sim \text{Unif}[0,1]$, *i.e.*,

$$\mu = \mathbb{E}_t[G(t)].$$

During training, this is estimated by the empirical mean over a batch:

$$\hat{\mu} = \frac{1}{n}\sum_{i=1}^{n} G(t_i).$$

If $t_i \sim$ i.i.d. $\text{Unif}[0,1]$ for $i = 1, \ldots, n$, then by standard results,

$$\text{Var}[\hat{\mu} \mid x_0] = \frac{1}{n}\sigma_t^2, \quad \text{where } \sigma_t^2 = \text{Var}_t[G(t)].$$

This corresponds to the Ⓑ term in the variance decomposition:

$$\mathbb{E}_{x_0}[\text{Var}_t(G)].$$

For stratified sampling, partition $[0,1]$ into $k$ equal-length intervals:

$$I_j = \left[\frac{j-1}{k}, \frac{j}{k}\right), \quad \text{for } j = 1, \ldots, k,$$

and sample $t_j \sim \text{Unif}(I_j)$ independently within each stratum. Then the estimator becomes

$$\hat{\mu} = \frac{1}{k}\sum_{j=1}^{k} G(t_j).$$

This remains an unbiased estimator of $\mu$, and its conditional variance is:

$$\mathrm{Var}[\hat{\mu} \mid x_0] = \frac{1}{k^2} \sum_{j=1}^{k} \mathrm{Var}_{t \in I_j}[G(t)] = \frac{1}{k} \sigma_w^2,$$

where we define the within-stratum variance:

$$\sigma_w^2 = \frac{1}{k} \sum_{j=1}^{k} \mathrm{Var}_{t \in I_j}[G(t)].$$

Decomposing the total variance $\sigma_t^2$ into within- and between-stratum variance gives:

$$\sigma_t^2 = \frac{n}{k}(\sigma_w^2 + \sigma_b^2), \quad \text{where } \sigma_b^2 = \mathrm{Var}_k[\mu_k], \quad \mu_k = \mathbb{E}_{t \in I_k}[G(t)].$$

Therefore, the variance of the stratified estimator satisfies:

$$\mathrm{Var}_{\mathrm{strat}}[\hat{\mu} \mid x_0] = \frac{1}{n}\sigma_t^2 - \frac{1}{k}\sigma_b^2,$$

which is smaller than the SRS variance:

$$\mathrm{Var}_{\mathrm{SRS}} = \frac{1}{n}\sigma_t^2.$$

The reduction is exactly

$$\frac{1}{k}\sigma_b^2.$$

**Practical Suggestion for Choosing $n$**

To choose an appropriate number of strata, we aim to balance two objectives:

1. Reduce the overall estimation variance (which favors using a larger number of strata $n$);
2. Ensure the estimation of within-stratum variance $\sigma_k^2$–denoted by $\hat{\sigma}_k^2$–is sufficiently stable (which favors having more samples per stratum, *i.e.*, $m = \mathrm{int}(B/n)$).

While setting $n = B$ results in highly fine-grained strata, it may lead to unstable estimates within each stratum due to too few samples. To balance this trade-off, we propose the following optimal number of strata:

$$n_{\mathrm{opt}} \propto \sqrt{B},$$

where $B$ is the total sampling budget, and $m = \mathrm{int}(B/n)$ denotes the number of samples per stratum.

### A.5.6 BIN-WISE EMA: VARIANCE REDUCTION & HYPERPARAMETER SELECTION

**Bin-Wise EMA Reduces Ⓑ as a Control-Variate Strategy** To estimate a random variable $X$ with mean $\mu_X = \mathbb{E}[X]$, we can use a control variate $Y$ with known mean $\mu_Y = \mathbb{E}[Y]$. Given $n$ i.i.d. samples $\{(X_i, Y_i)\}_{i=1}^{n}$, the control variate estimator

$$\mu_{CV} = \bar{X} - \frac{\mathrm{Cov}(X,Y)}{\sigma_Y^2}\bar{Y}$$

reduces the variance from $\frac{\sigma_X^2}{n}$ to $\frac{\sigma_X^2}{n}(1 - \rho^2)$, where $\rho = \frac{\mathrm{Cov}(X,Y)}{\sigma_X \sigma_Y}$. Importantly, this form of estimator gives an unbiased estimate of loss gradients.

In our context, $X$ represents the loss gradient for the current batch, while $Y$ is the exponential moving average $b_k$ of past gradients within the same $t$-bin. Assuming strong heteroscedasticity over $t \in [0,1]$, partitioning $t$ into bins and applying EMA within each bin maximizes $|\rho|$, thus achieving significant variance reduction. Several alternative control strategies are listed in Table 3. Among these, we adopt bin-wise EMA for its strong variance reduction under heteroscedastic conditions and minimal implementation burden, which also helps prevent overfitting.

| Variant | Assumptions | Tuning and Resources | Characteristics |
|---------|-------------|----------------------|-----------------|
| Spline | Smooth $\ell(t)$, no sharp changes | Requires degree/node selection; low memory usage | Captures trends well but sensitive to under-/over-fitting |
| Kernel | Flexible; handles arbitrary heteroscedasticity | Only bandwidth $h$; stores $O(1/h)$ samples; higher memory cost than bin-wise EMA | Most robust; no shape assumptions; tuning and implementation are more complex |

Table 3: Comparison of spline and kernel variants

**Choose the Number of Bins in EMA: A Mathematical Perspective**   Suppose we have $m$ data points and choose $b$ bins; then each bin will contain approximately $m/b$ samples to estimate the piecewise-constant value of $b(t)$ over an interval of width $1/b$.

We use MSE optimization analysis to determine a principled choice of $b$ given $m$. Specifically, we aim to estimate the function

$$g(t) = \mathbb{E}_{x_t}[\ell_\theta(x_0, t, x_t) \mid x_0, t]$$

on $[0, 1]$ by a piecewise-constant approximation $g_b(t)$ with $b$ equal-width bins of size $\Delta = 1/b$, using the sample average within each bin.

The total mean squared error is decomposed as

$$\text{MSE}(B) = \underbrace{\mathbb{E}[(g_b(t) - g(t))^2]}_{\text{Bias}^2} + \underbrace{\text{Var}(g_b(t))}_{\text{Variance}}.$$

**Bias Term**   Assuming $g(t)$ is twice differentiable within each bin, we can apply Taylor approximation:

$$\text{Bias}^2 \approx C_1 \Delta^4 = \frac{C_1}{b^4},$$

where $C_1 \propto \max|g''(t)|$.

**Variance Term**   Each bin contains $n = m/b$ samples, hence:

$$\text{Var} \approx \frac{C_2}{n} = \frac{C_2 b}{m},$$

where $C_2 \propto \text{Var}_{\text{mask}}[\ell]$.

**Total MSE**   Combining both terms:

$$\text{MSE}(B) \approx \frac{C_1}{b^4} + \frac{C_2 b}{m}.$$

**Optimal $b$**   Taking the derivative *w.r.t.* $b$ and setting it to zero:

$$\frac{d}{db}\text{MSE}(B) = -4C_1 b^{-5} + \frac{C_2}{m} = 0 \quad \Rightarrow \quad b^* = \left(\frac{4C_1 m}{C_2}\right)^{1/5} \propto m^{1/5}.$$

**Selection for $\eta$**   For $\eta$, it's useful to think in terms of the e-folding time $k = -\frac{1}{\ln(1-\eta)} \approx \frac{1}{\eta}$, so that each EMA bin effectively averages over the past $k \times$ (batch size per GPU) losses. Larger $\eta$ adapts faster but use fewer past samples to estimate $b(t)$; smaller $\eta$ is stabler but can cause the EMA to lag (and suffer from domain-shift issues). In our experiments, we set:

$$m \times \left(\frac{1}{\eta} \times \text{batch size per GPU}\right) \approx 0.1 \times (\text{train data size}).$$

### A.6   METHOD COMBINATION

One might wonder whether all methods proposed in this work could be combined into a single ultra-strong approach. While such a combination may sound attractive, it is not a straightforward task: different methods rely on distinct assumptions, and naive aggregation can easily introduce redundancy or bias:

1. **Different Assumptions.** Each method relies on specific assumptions. For example, the effectiveness of P-POTS depends on accurate modeling of $g(t)$ and $v(t)$, while Bin-Wise EMA requires heteroskedasticity of the loss $l$ with respect to $t$. Blindly combining them risks violating these assumptions.

2. **Redundancy.** Each method in this work is already effective on its own. If two methods target the same source, the first may remove most of the variance (*e.g.*, 80%), while the second may introduce estimation bias that offset any residual variance reduction, resulting in worse performance than using a single method.

Instead, we emphasize the synergy between P-POTS and MIRROR, which our experiments confirm to be stable. In practice, we recommend P-POTS+MIRROR for maximal performance gains, and P-POTS alone for cost efficiency. Other techniques may still be preferable in specific contexts due to their simplicity or ease of implementation. Rather than combining everything, we provide guidance below on how to select among them.

### A.6.1 STRATIFIED SAMPLING VS EMA

Both methods rely on the assumption that the loss $l$ is heteroskedastic in $t$: after averaging over $x_0$ and $x_t$, different values of $t$ yield losses of different magnitudes. This generally holds, since larger $t$ masks more tokens, making reconstruction harder and the loss higher, as shown in Figure 5a.

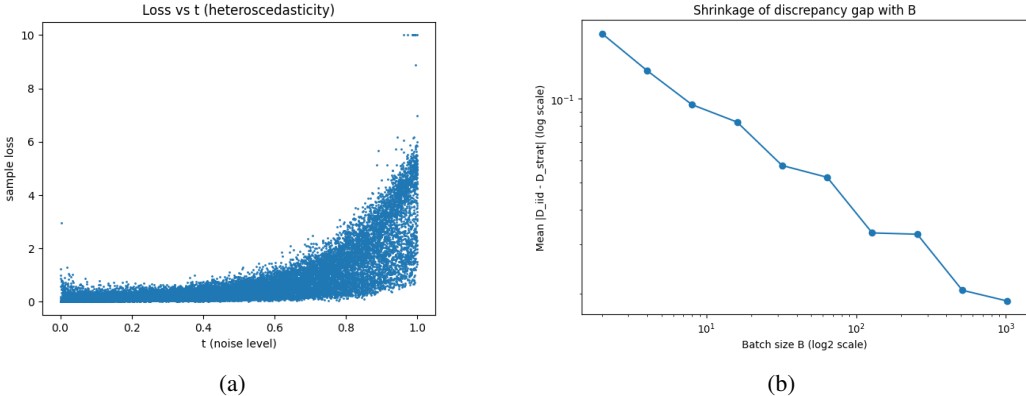

(a)  (b)

Figure 5: The left panel shows the heteroscedasticity of the $t$-loss, and the right panel illustrates the shrinking difference between IID and stratified sampling as batch size $B$ increases, where we use the KS statistic to measure their maximum deviation.

However, the two methods reduce Ⓑ in fundamentally different ways:

- Batch Size: Stratified sampling enforces fixed proportions of $t$-values in each batch, which is particularly beneficial when the batch size is small. As the batch size grows, its advantage over simple independent sampling diminishes, as shown in Figure 5b. By contrast, EMA accumulates information continuously across training; its effectiveness depends on the choice of $m$ and $\eta$, but not on batch size.

- Theoretical Limit: Stratified sampling cannot fully eliminate variance, as within-stratum variance always remains. In contrast, EMA can, in principle, reduce Ⓑ to zero if the control variate is sufficiently correlated.

**Note.** We do not recommend combining EMA with stratified sampling. Each method is effective on its own, and their joint use may introduce estimation bias that offset any residual variance reduction.

In practice, we suggest using stratified sampling as the default strategy for reducing Ⓑ, and switching to EMA only when the batch size is sufficiently large.

### A.6.2   ISAD VS SYRM

These methods are dataset-specific. We recommend choosing based on the dataset type: ISAD is designed for QA datasets with explicit answer delimiters, while SyRM is designed for datasets containing syntax tokens in prompts.

## A.7   EXPERIMENTAL DETAILS

### A.7.1   HYPERPARAMETER SETTINGS

We detail here the hyperparameters used for training and inference of masked diffusion models (MDMs) and auto-regressive models (ARMs).

**Chain-of-Thought (CoT) Reasoning.**

| Dataset | CoT Handling |
|---|---|
| All except text-to-image-2M & HiTab | Responses include chain-of-thoughts |
| HiTab | No CoT provided; generated via rejection sampling |

**Train/Validation/Test Splits.**

| Dataset | Train/Validation Split | Test Split |
|---|---|---|
| OpenScience | 5,000 samples, 9:1 (4,500/500) | 5001st–6000th examples |
| GSM8K | Official train split | Official test split |
| HiTab | Official train split | Official test split |
| text-to-image-2M | 5,000 samples, 9:1 (4,500/500) | 5001st–5100th examples |

text-to-image-2M consists of two subsets, 1024_10k and 512_2M. We sample 2,500 examples from each subset to construct the training and validation splits, and 500 examples from each subset to form the test split.

**General Hyperparameter.**

| Parameter | Value |
|---|---|
| Learning rate | $5 \times 10^{-5}$ |
| Scheduler | Linear |
| Global batch size | 32 |
| Epochs | 5 |
| Max sequence length | 4096 |
| Warmup steps | 0 |

Note that after constructing the training data as described in **Train/Validation/Test Splits.**, we further filter out samples with a total sequence length (prompt + response) exceeding 4096, matching the fixed context length used in the pretraining stage of LLaDA. As a result, the final dataset may be smaller than the original.

**Method-Specific Hyperparameters (MDMs Only).**

| Technique | Setting |
|---|---|
| Importance sampling | $\Delta = 0.2$ |
| Stratified sampler | $\#\text{strata} = \lceil\sqrt{\text{batch size}}\rceil = 6$ |
| EMA bins | 10 |
| EMA learning rate | 0.01 |

**Checkpoint Selection.**

| Criterion | Setting |
|---|---|
| Evaluation strategy | Early stopping (patience = 3) |

**Random Seeds (MDMs).**

| Seeds used | 42, 731, 20231 |
|---|---|

**How Data Cleaned in HiTab**

Raw answers are extracted from `\boxed{}` delimiters, then cleaned by removing surrounding whitespace, unwrapping `\text{...}` blocks, converting escaped percent signs `\%` to %, deleting backslashes and trimming a trailing period

**Inference Settings for LLaDA.**

We set the decoding temperature to $0$ for all ARMs and MDMs, ensuring that variance in downstream task performance originates solely from training.

| Dataset | Train Data Response Max Length | ARMs | MDMs |
|---|---|---|---|
| GSM8K | 444 | 512 | 128-128-32 |
| HiTab | 3571 | 3968 | 512-256-16 |
| Open | 4605 | 5120 | 256-256-256 |

The ARM column indicates the maximum number of new tokens, while the MDM column specifies generation length-steps-block length, as we adopt block diffusion for inference (Arriola et al., 2025).

For ARMs, we determine the maximum new tokens as:

$$\text{max new tokens} = \lceil \text{response max length} \times (1 + s) \rceil_{64},$$

where $s$ is a safety factor (set to $0.1$), and the result is upscaled to the nearest multiple of $64$.

For MDMs, hyperparameters are selected according to dataset type.

**Note.** As shown in the table, ARMs require significantly longer generation lengths at inference. This arises from their auto-regressive nature: given a budget of max new tokens, they auto-regressively generate token by token, without explicit awareness of how many tokens are ultimately needed. In contrast, MDMs use substantially fewer tokens-even shorter than the training responses-yet still produce accurate reasoning without truncation. This efficiency arises because MDMs predict all masked tokens in parallel and are constrained by the generation length. For example, with `gen length = 512`, the model predicts all 512 tokens for the masked input and iteratively applies the remasking–masking scheduler until completion. By design, MDMs avoid unnecessarily verbose outputs, which we argue constitutes another advantage of MDMs over ARMs.

**Inference Settings for MMaDA.**

| Dataset | generation length | steps | block_length | temperature | mask schedule |
|---|---|---|---|---|---|
| text-to-image-2M | 1024 | 15 | 1024 | 0 | cosine |

### A.7.2 HOW TRAINING VARIANCE IS ESTIMATED

During training, we maintain an online estimator of the per-sample loss variance using importance-weighted statistics. Since losses decrease during training, this metric captures both fluctuations and the overall downward trend, it is an approximate empirical proxy for Eq. 2

For each batch, the per-token (or per-sample) losses $L$ are detached, flattened, and reweighted by the corresponding importance weights $w$. We then update three running sums: the weighted mean of losses ($S_1$), the weighted mean of squared losses ($S_2$), and the weighted mean of squared weighted losses ($S_{12}$). These sufficient statistics allow us to compute unbiased estimates of the second mo-

ment $m_2$, the squared mean $\mu^2$, and finally the unbiased variance estimator

$$\widehat{\text{Var}}[L] \;=\; m_2 - \mu^2, \tag{23}$$

where $m_2 = \frac{S_2}{n}$ and $\mu^2 = \frac{S_1^2 - S_{12}}{n(n-1)}$. Here, $n$ denotes the total number of accumulated samples across updates. If importance sampling on $t$ is enabled, the weights $w$ are drawn from the fitted distribution $p(t)$ to ensure unbiased estimation; otherwise, unit weights are used.

At the end of training, the accumulated statistics are queried once to produce the final unbiased variance estimate. These values are then substituted into Eq. 23 to obtain the final estimate of training variance.

Intuitively, the use of an unbiased estimator is necessary because the naive sample variance can be downward biased when computed incrementally with importance weights. By correcting for this bias, our estimate better reflects the true variance of the training process, independent of batch size or logging frequency.

## A.8 COMPLETE EXPERIMENTAL RESULTS

### A.8.1 PER-RUN EXPERIMENTAL RESULTS ON LLADA

| Dataset / Method | Seed 42 | | | Seed 731 | | | Seed 20231 | | |
|---|---|---|---|---|---|---|---|---|---|
| | Perf | Time (h) | Var | Perf | Time (h) | Var | Perf | Time (h) | Var |
| **OpenScience** | | | | | | | | | |
| Baseline | 43.57 | 11.29 | 2.1404 | 46.50 | 11.32 | 2.0614 | 46.50 | 11.40 | 2.0743 |
| P-POTS+Mirror | 50.80 | 14.70 | 1.5550 | 53.80 | 14.07 | 1.5672 | 53.00 | 14.00 | 1.5776 |
| P-POTS | 45.34 | 13.63 | 1.6319 | 47.60 | 12.98 | 1.6417 | 47.47 | 12.88 | 1.6622 |
| Mirror | 45.63 | 12.46 | 2.0800 | 47.25 | 12.47 | 2.0392 | 46.25 | 12.47 | 2.0205 |
| **GSM8K** | | | | | | | | | |
| Baseline | 50.64 | 1.55 | 0.2526 | 53.22 | 1.62 | 0.2524 | 53.68 | 1.61 | 0.2673 |
| P-POTS+Mirror | 62.02 | 3.64 | 0.1906 | 58.61 | 3.73 | 0.1949 | 60.96 | 3.88 | 0.1922 |
| P-POTS | 59.59 | 1.67 | 0.1726 | 59.36 | 1.77 | 0.1715 | 56.79 | 1.67 | 0.1722 |
| Mirror | 56.56 | 3.69 | 0.3167 | 51.40 | 3.37 | 0.3042 | 53.15 | 3.38 | 0.3162 |
| **HiTab** | | | | | | | | | |
| Baseline | 52.96 | 8.22 | 1.0063 | 60.43 | 8.22 | 1.0116 | 62.64 | 8.26 | 1.0173 |
| P-POTS+Mirror | 66.02 | 17.80 | 0.9903 | 66.67 | 16.78 | 0.9892 | 68.62 | 16.83 | 0.9847 |
| P-POTS | 60.56 | 10.19 | 0.7498 | 64.74 | 10.66 | 0.7536 | 58.80 | 10.99 | 0.7503 |
| Mirror | 64.72 | 16.08 | 1.3052 | 65.04 | 16.27 | 1.3282 | 63.68 | 16.29 | 1.3480 |

Table 4: Complete results –performance, training time (in hours), and variance – across datasets (OpenScience, GSM8K, HiTab).

Compared to only reporting mean and standard deviations across runs, this complete results table provides a more direct view of training instability: under standard training, the final performance can vary substantially, *e.g.*, from 52.96% to 62.64% on HiTab. In contrast, our method achieves more consistent outcomes across runs, demonstrating greater reliability.

Table 4 and 5 show that methods with lower reported training variance are generally associated with better downstream performance. Interestingly, MIRROR sometimes reports higher variance than the baseline, yet achieves more stable results in evaluation. Since our variance measure serves only as a proxy that captures both the downward trend and short-term fluctuations of training losses, we conjecture that the stronger overall loss reduction under MIRROR may interact with this proxy in a way that inflates the reported variance, even while true training stability improves.

Table 5: Average training variance with training hours across datasets (All experiments were trained on 2 H100 GPUs; details on training variance approximation provided in A.7.2)

| | OpenScience | GSM8K | HiTab |
|---|---|---|---|
| Baseline | 2.0920 (11.33) | 0.7723 (1.59) | 3.0352 (8.23) |
| P-POTS+MIRROR | 1.5666 (14.26) | 0.5777 (3.75) | 2.9642 (17.13) |
| P-POTS | 1.6453 (13.17) | 0.5163 (1.71) | 2.2537 (10.61) |
| MIRROR | 2.0466 (12.47) | 0.9371 (3.48) | 3.9814 (16.21) |
| ISAD | 2.1480 (11.48) | 0.2525 (1.61) | 1.0016 (10.51) |
| SyRM | – | – | 0.9233 (8.23) |
| StraTS | 2.0851 (11.46) | 0.2568 (1.59) | 1.0083 (10.19) |
| EMA | 2.1450 (11.46) | 0.2535 (1.62) | 1.0018 (8.40) |

### A.8.2 PER-RUN EXPERIMENTAL RESULTS ON MMADA

| Dataset / Method | Seed 42 | | | Seed 731 | | | Seed 20231 | | |
|---|---|---|---|---|---|---|---|---|---|
| | Perf | Time (h) | Var | Perf | Time (h) | Var | Perf | Time (h) | Var |
| Baseline | 28.61 | 3.21 | 1.4149 | 34.28 | 3.09 | 1.4245 | 31.19 | 3.31 | 1.3848 |
| MIRROR | 34.88 | 6.88 | 1.0308 | 33.77 | 6.13 | 1.0808 | 33.57 | 6.74 | 1.0458 |
| **text-to-image-2M EPR** | | | | | | | | | |
| P-POTS | 31.51 | 3.28 | 2.6203 | 30.02 | 3.33 | 2.6150 | 33.41 | 3.37 | 2.6236 |
| P-POTS+MIRROR | 35.27 | 6.97 | 2.4044 | 34.10 | 6.79 | 2.3504 | 34.39 | 6.87 | 2.3370 |
| **text-to-image-2M polynomial (degree 7)** | | | | | | | | | |
| P-POTS | 30.00 | 3.28 | 2.8395 | 33.35 | 3.27 | 2.8398 | 28.66 | 3.56 | 2.6187 |
| P-POTS+MIRROR | 34.51 | 5.96 | 2.3538 | 33.78 | 6.43 | 2.7319 | 34.78 | 6.71 | 2.3199 |

Table 6: Performance (Perf), training time (in hours), and variance (Var) on text-to-image-2M-EPR and text-to-image-2M-poly7 across methods and random seeds (42, 731, 20231).

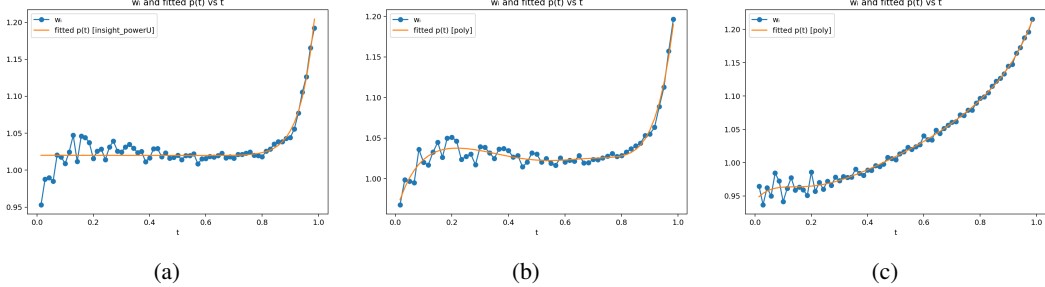

| (a) | (b) | (c) |
|---|---|---|

Figure 6: Shape of $p(t)$ under different situations: (a) shape of empirical and fitted $p(t)$ by EPR before training; (b) by polynomial of degree 7 before training; (c) fitted by EPR after training for one epoch using polynomial (degree 7). These functions are fitted by minimizing the KL divergence, so only their shapes are matched; they are later renormalized in training to yield valid PDFs.

As shown in Figure 6, even when the model is trained with the S-shaped distribution, after one epoch it quickly reverts back to the U-shape. This suggests that the EPR form is a stable and persistent structure across datasets.

### A.9 EXTENSION TO REINFORCEMENT LEARNING

Applying standard RL algorithms such as PPO and GRPO to MDMs is challenging, because it requires computing log-likelihoods over transitions between consecutive states. In ARMs, states are token sequences. Since tokens are generated one by one, by the chain rule $P(x_{1:n}) = \prod_{i=1}^{n} P(x_i \mid x_{<i})$, the transition probability for each token $x_i$ is naturally $p_\theta(x_i \mid x_{<i})$. In contrast, MDMs generate tokens in parallel, which induces complex marginalization between intermediate states $x_{t-1}$ and $x_t$. Training stability is further challenged by token-level rewards derived from high-variance sequence returns.

---

**Algorithm 1** UniGRPO Policy Gradient Optimization

---

**Require:** Reference model $\pi_{\text{ref}}$, prompt distribution $\mathcal{D}$, number of completions per prompt $G$, number of inner updates $\mu$, diffusion steps $T$

1: Initialize policy $\pi_\theta \leftarrow \pi_{\text{ref}}$
2: **while** not converged **do**
3:     $\pi_{\text{old}} \leftarrow \pi_\theta$
4:     Sample a prompt $q \sim \mathcal{D}$
5:     Sample $G$ completions $o_i \sim \pi_{\text{old}}(\cdot \mid q)$ for $i \in [G]$
6:     For each $o_i$, compute reward $r_i$ and advantage $A_i^{(k)}(\pi_{\text{old}})$
7:     Sample a starting timestep $t_0 \sim \mathcal{U}(0, T-1)$
8:     Generate $\mu - 1$ uniformly spaced timesteps $t_1, \ldots, t_{\mu-1}$ from $[t_0, T]$
9:     **for** gradient update iterations $n = 1$ to $\mu$ **do**
10:        **if** $n = 1$ **then**
11:            Sample a starting mask ratio $r_1 \sim \mathcal{U}(0, 1)$ and compute initial timestep $t_1 = \lfloor r_1 T \rfloor$
12:        **else**
13:            Compute $t_n = \left\lfloor \frac{n-1}{\mu-1}(T - t_1) + t_1 \right\rfloor$
14:        **end if**
15:        Construct input $(q, \text{masked } o_i)$ using timestep $t_n$ (with $q$ always unmasked)
16:        For $\pi_\theta, \pi_{\text{old}}, \pi_{\text{ref}}$, estimate log-probabilities of masked tokens in $o_i$ at $t_n$
17:        Compute UniGRPO objective 24 and update $\pi_\theta$ via gradient descent
18:     **end for**
19: **end while**
20: **return** $\pi_\theta$

---

Our methods can be naturally extended to these problems. As an example, consider UniGRPO (Yang et al., 2025) (Algorithm 1), whose training objective is defined as

$$\mathcal{J}_{\text{UniGRPO}}(\theta) = \mathbb{E}_{(q,a)\sim\mathcal{D},\,\{o_i\}_{i=1}^G \sim \pi_{\theta_{\text{old}}}(\cdot|q),\,\{p_i \in [0,1]\}_{i=1}^G}$$
$$\left[ \frac{1}{G} \sum_{i=1}^G \frac{1}{|o_i|} \sum_{t=1}^{|o_i|} \Big( \min\big(r'_{i,t}(\theta)\,\hat{A}_{i,t},\,\text{clip}(r'_{i,t}(\theta),\, 1-\varepsilon,\, 1+\varepsilon)\,\hat{A}_{i,t}\big) - \beta D_{\text{KL}}\big(\pi'_\theta \,\|\, \pi'_{\text{ref}}\big) \Big) \right]. \quad (24)$$

We can incorporate our core methods through the following modifications:

- P-POTS: Replace lines 7–8 with sampling $\mu$ timesteps from $p^*(t)$, sorted as $t_0, \ldots, t_{\mu-1}$.

- MIRROR: At line 15, construct complementary masks $o_i^{(1)}$, $o_i^{(2)}$ for each $t_n$, and average their log-probabilities before updating $\pi_\theta$.

At present, this integration is a conjecture; confirming its effectiveness requires empirical validation, which we leave as future work.

## A.10    FUTURE WORK

While our current results highlight both the importance of variance reduction and the effectiveness of our proposed methods, several open directions remain. First, we aim to deepen our understanding of the $p$-drift problem and to validate the effectiveness of periodically re-estimating or gradually down-weighting $p(t)$, as suggested in §3.1.1, which so far remains conjectural. Second, since reinforcement learning on MDMs face the difficulty of likelihood evaluation, it is worthwhile to explore whether our methods can also reduce estimation variance in this setting (see A.9).

## A.11    CASE STUDY

### A.11.1    CASE STUDY: IMAGE GENERATION

Below are images generated by MMaDA-8B-MixCoT trained with P-POTS+MIRROR (left) and standard method (right).

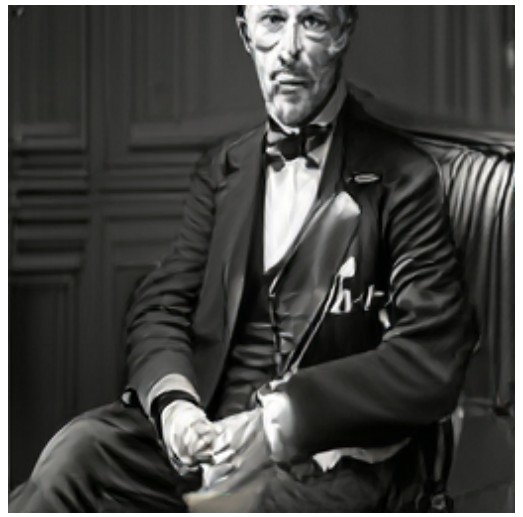
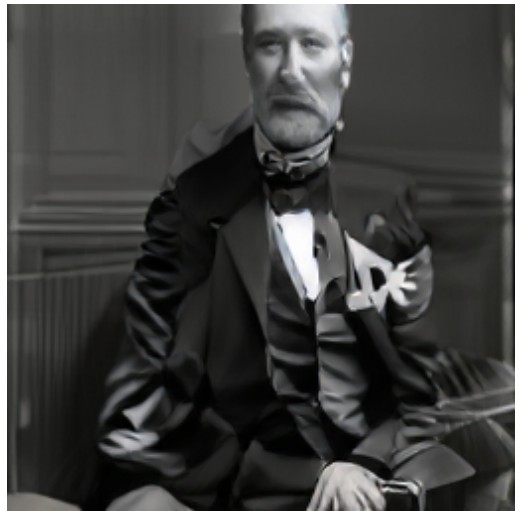

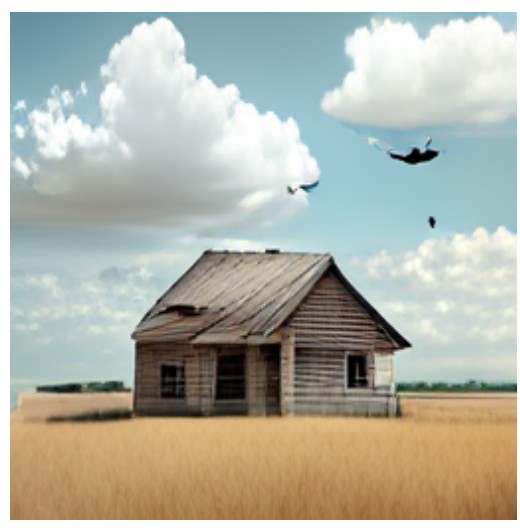
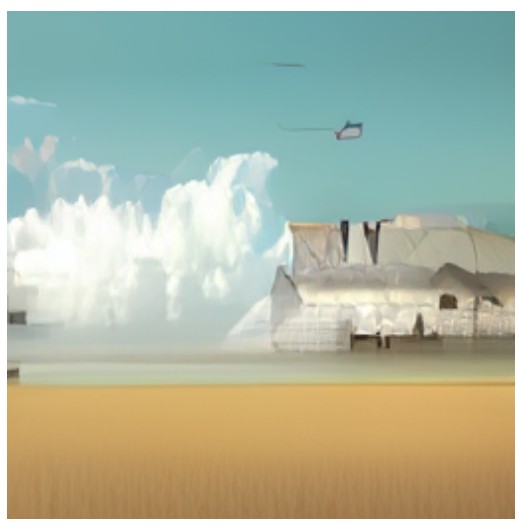

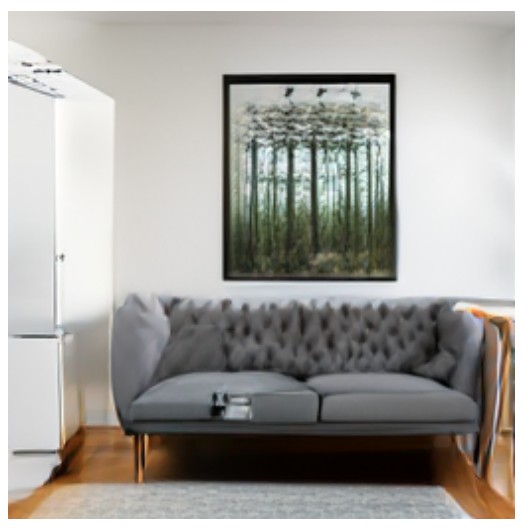
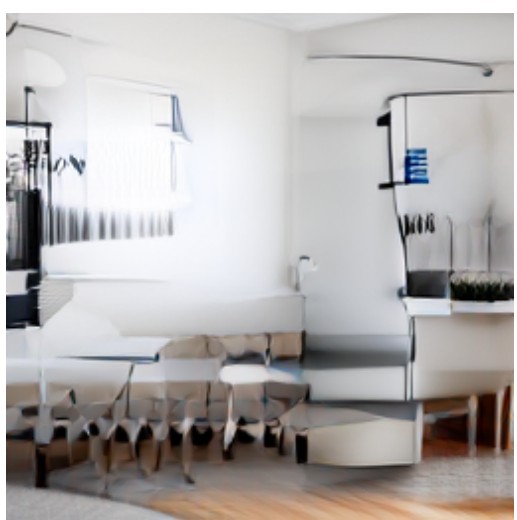

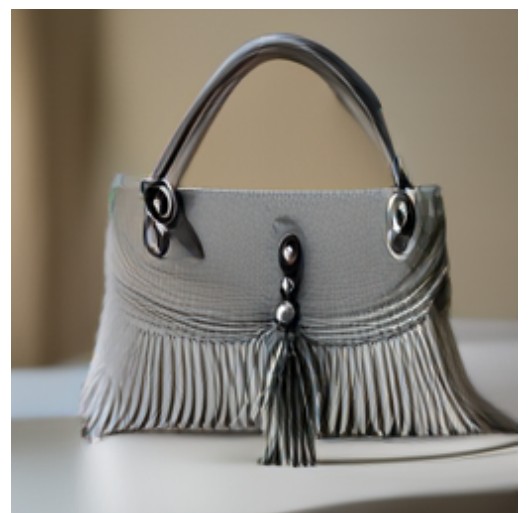
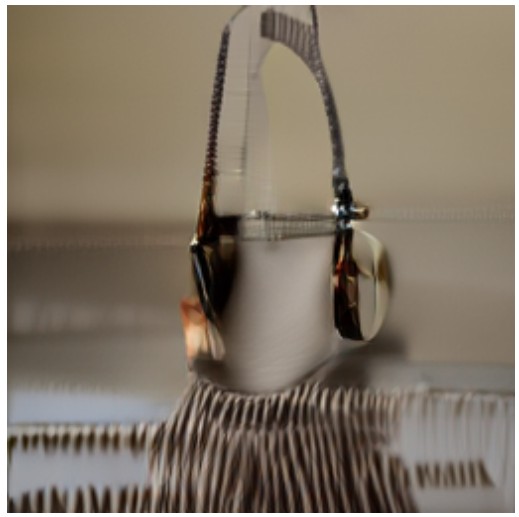

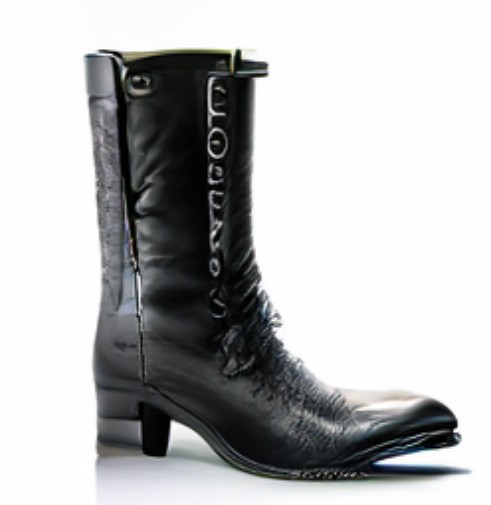
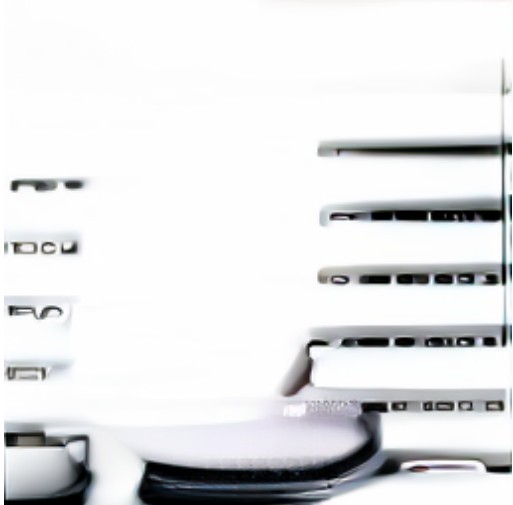

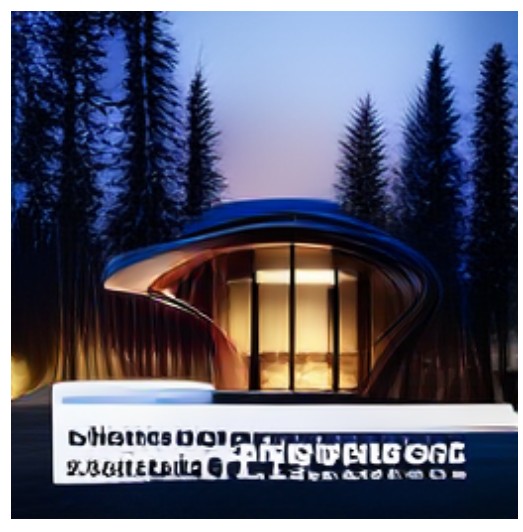
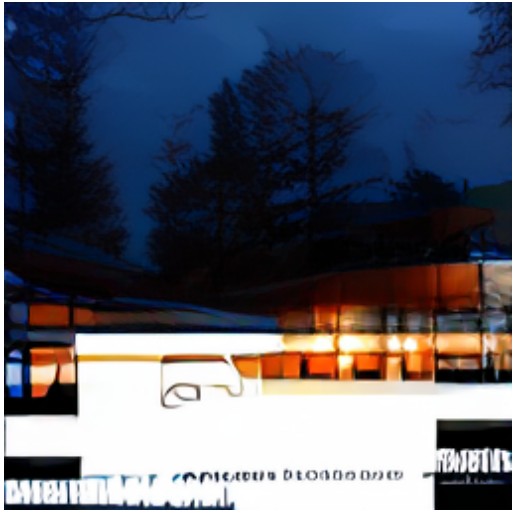

