# OpenReview forum: "Bringing Stability to Diffusion: Decomposing and Reducing Variance of Training Masked Diffusion Models"
_ICLR.cc/2026/Conference — ICLR 2026 Poster_

### Official Review · Reviewer_J4Ut · 2025-10-30

**Soundness:** 4
**Presentation:** 3
**Contribution:** 4
**Rating:** 8
**Confidence:** 2

**Summary:**

This paper tackles the critical problem of high training variance in Masked Diffusion Models (MDMs), which causes optimization instability and performance degradation compared to Auto-Regressive Models (ARMs). The paper derives the first systematic decomposition of MDM training variance, identifying three distinct sources: A) Masking Pattern Noise, B) Masking Ratio Noise, and C) Data Noise. Based on this decomposition, the authors propose six methods, including two core techniques: P-POTS and MIRROR.  Experiments on complex reasoning and multimodal tasks show that the proposed methods, particularly the combination "P-POTS+MIRROR," dramatically improve performance (e.g., +7-8% accuracy on GSM8K) and, critically, reduce run-to-run performance variability.

**Strengths:**

1. This paper provides a clean, intuitive, and theoretically sound framework for understanding why MDMs are unstable.
2. The proposed core methods (P-POTS and MIRROR) are well-motivated and directly derived from the theoretical analysis.
3. The significant improvement in the MDM's performance makes the proposed method not only theoretically sound but also practically effective.

**Weaknesses:**

This is a strong paper without major weaknesses. One suggestion on the presentation is that although this paper proposes two core methods to reduce the instability, it introduces six methods in total. I think this may harm the readability of this paper and distract the reader from understanding the core contribution. Therefore, the author may reconsider the paper structure.

**Questions:**

1. Based on my understanding, LLaDA and Dream seem to use slightly different formulations to train MDM. However, this paper only conducts experiments on LLaDA. Thus, are the proposed methods applicable to other baselines, such as Dream?
2. This paper discusses two types of LM, MDM and ARM, where MDM is fully bidirectional. How about the combined version, such as BlockDiffusion[1]? Does the author have any insights on this case, such as the training variance or whether proposed methods are still applicable?
3. Though experimental results are quite impressive, I still recommend that the author provide train-from-scratch results in the future since the training variance in pretraining might be more salient.


[1] Block Diffusion: Interpolating Between Autoregressive and Diffusion Language Models, in ICLR 2025

---

> ### Author Response · Authors · 2025-11-22
>
> Thank you very much for your thoughtful feedback. We are delighted that you found our theoretical framework for analyzing MDM instability both intuitive and sound, and that you considered proposed P-POTS and MIRROR methods well-motivated and practically effective. Let us answer your questions below.
>
> ---
>
> ## 1) Does the proposed method apply to other baselines like Dream-7B?
>
> This paper conduct experiments on two MDMs: (1) **LLaDA-8B-Instruct** and (2) **MMaDA-8B-MixCoT**. We did attempt to include additional MDM baselines such as Dream-7B, but had faced two significant challenges:
>
> 1. **Limited training frameworks.** No open-sourced pretrained MDMs (even including LLaDA and MMaDA) released training code when writing this paper; they only release inference code. The experimental results in our paper rely on training code we implemented from scratch following the MDM training logic (3k+ lines of codes), which made extending to many other baselines non-trivial.
>
> 2. **Computational constraints.** A central claim of our paper is that our method yields **both** a better optimum and stable convergence across runs. To validate this, we repeated full training for 8B-scaled models three times (seeds 42, 731, 20231), tripling compute. Overall, this required about 500 H100 GPU-hours for training and 330 H100 GPU-hours for inference.
>
> Going forward, we will actively track newly released MDM training code and checkpoints as an important direction for future work.
>
> ---
>
> ## 2) Does the proposed method apply to Block Diffusion?
>
> Yes. Our variance decomposition and the Pareto-optimal sampler (P-POTS) are derived without any hard assumptions, so the optimality argument still holds. Since Block Diffusion is autoregressive across blocks and diffusion within blocks, the AR component introduces no extra masking noise; therefore, we can directly apply our method to the within-block diffusion part.
>
> In particular, the Block Diffusion paper also identifies high training variance in MDMs as a key challenge. To reduce variance, they propose a **clipped noise scheduler**: it selects a best clipped interval $[a,b]\subset[0,1]$ from e.g. $12$ candidates and then samples $t$ uniformly from this chosen interval. Empirically, they find $[0.45, 0.95]$ to work best and show that the clipped scheduler performs the best in both NLL and variance over most standard schedulers. This validates the importance of variance-reduction in block diffusion architectures.
>
> Importantly, our paper includes their clipped noise scheduler as a baseline. In our paper, we repeat **training (not inference)** for three times (seeds: 42, 731, 20231), holding model, data, hyper-parameters all fixed. We then evaluate the trained models by **greedy decoding** to ensure that all difference across runs comes from training, not inference. The results are summarized as follows:
>
> | Model (Method) | OpenScience | GSM8K | HiTab |
> |---|---:|---:|---:|
> | **ARMs** ||||
> | Llama-3.1-8B-Instruct | 44.95 | 47.60 | 65.02 |
> |  | 47.07 | 46.40 | 66.28 |
> |  | 47.76 | 49.10 | 64.88 |
> | **MDMs** ||||
> | LLaDA-8B-Instruct (Baseline) | 43.57 | 50.64 | 52.96 |
> |  | 46.50 | 53.22 | 60.43 |
> |  | 46.50 | 53.68 | 62.64 |
> | LLaDA-8B-Instruct (Clipped) | 47.90 | 55.57 | 58.90 |
> |  | 49.20 | 54.60 | 60.80 |
> |  | 50.90 | 58.38 | 62.20 |
> | LLaDA-8B-Instruct (P-POTS+Mirror) | 50.80 | 62.02 | 66.02 |
> |  | 53.80 | 58.61 | 66.67 |
> |  | 53.00 | 60.96 | 68.62 |
>
> As shown in the table, our method consistently achieves significant better results across runs compared to both standard method and clipped scheduler. This agrees with the expected Pareto-optimal frontier: PPOTS is not heuristic and is derived as the variance minimiser, so no other $t$-sampler should achieve lower loss variance under the same estimated objective. In almost all cases, even their **best** performance remains well below PPOTS+MIRROR's **worst** performance. Beyond improved average quality of trained models, our method reduces the run-to-run variability to almost the same level of ARMs (Llama-3.1-8B-Instruct).
>
> In short, their work supports the effectiveness of variance reduction on Block Diffusion, while our results show that P-POTS+MIRROR achieves stronger variance reduction and higher final performance. Hence, we believe, both theoretically and empirically, that proposed methods will remain effective in block diffusion.
>
> ## 3) Train-from-Scratch Results and Paper Structure
>
> We agree that train-from-scratch results would further strengthen the paper, and we plan to include them as future work as computational resources allow. We also appreciate your suggestion regarding paper structure. In the revision, we will move some details of the auxiliary techniques to the appendix to improve readability and focus.
>
> ---
>
> In conclusion, we sincerely thank you again for your helpful feedback. We hope this reply addresses your uncertainty, and we are very happy to answer any further questions you may have.

---

> ### Comment · Reviewer_J4Ut · 2025-11-26
>
> Thanks for the detailed response. My concerns have been mostly addressed.

---

### Official Review · Reviewer_WDaB · 2025-10-30

**Soundness:** 3
**Presentation:** 2
**Contribution:** 2
**Rating:** 6
**Confidence:** 4

**Summary:**

This paper try to decompose the MDM's training noise. The noise is divided into three parts: data noise and two about masking.
The core analyses are about inter-group and inner-group decoupling by the law of the total variance. Based on the aforementioned analyses, two main approaches are proposed, P-POTS and MIRROR.

**Strengths:**

- Intuitive and sharp derivation of the theorem provides a mathemetically elegant  and practical explanation.
- Numerical pre-experiments seems robust and adheren to the expected Pareto frontier.

**Weaknesses:**

- Limited experiments about generalization and comparison. The ablaition experiments are mixed in the table of comparison. The included MDM baselines are too limited.
- Parts of the error bars are missing, and meanwhile, the error bars reported are too large to convince audience that the methods are consistently performing well as it's tested in the main table.

**Questions:**

- Selected benchmarks are all about QA reasoning tasks. How about general QA tasks and knowledge-intensive benchmarks, e.g., Graph QA, OCR detection?

---

> ### Author Response · Authors · 2025-11-22
>
> Thank you very much for your helpful feedback. We are pleased that you found our theoretical derivation intuitive and mathematically elegant. We also appreciate your comments on presentation and contribution, which we address in detail below.
>
> ---
>
>
>
> ## 1) The included MDM baselines are too limited.
>
>
> In this paper, we conduct experiments on two MDMs: (1) **LLaDA-8B-Instruct** and (2) **MMaDA-8B-MixCoT**. We agree that broader coverage would be valuable, but we had faced significant difficulties when writing this paper:
>
> 1. **Limited Code.** No pretrained MDMs (including LLaDA and MMaDA) open-sourced their training code when writing this paper; they only provide inference code. In our paper, we re-implement training code from scratch (≈3k+ lines of code). This made it non-trivial to extend experiments to many additional MDM baselines.
>
> 2. **Computational cost of variance evaluation.** To demonstrate that our methods improve performance consistently, we repeat **full training (not inference) ** of 8B-scale models three times under different random seeds (42, 731, 20231), which triples both training and inference cost (in total, our study required about 500 H100 GPU-hours for training and 330 H100 GPU-hours for inference).
>
> 3. **Limited Pretrained MDMs.** Compared to ARMs, the number of publicly available pretrained MDMs is still very small. This significantly constrains baseline diversity at comparable scale.
>
> We will continue to track future MDM training-code releases and view expanding to more families as an important direction for future work.
>
> ---
>
> ## 2) The error bars reported are too large to convince audience that the methods are consistently performing well as it's tested in the main table.
>
> Thank you for raising this point -- we realize our presentation may not have been clear enough. The values after “±” in the main table are **standard deviations**, not error bars. Concretely, for each setting we repeat **training (not inference)** three times with the same model, data, and hyperparameters, differing only in random seeds (42, 731, 20231). We then evaluate the trained models using **greedy decoding** to ensure that performance differences across runs come solely from training, not inference. The run-to-run results are summarized below:
>
> | Model (Method) | OpenScience | GSM8K | HiTab |
> |---|---:|---:|---:|
> | **ARMs** ||||
> | Llama-3.1-8B-Instruct | 44.95 | 47.60 | 65.02 |
> |  | 47.07 | 46.40 | 66.28 |
> |  | 47.76 | 49.10 | 64.88 |
> | **MDMs** ||||
> | LLaDA-8B-Instruct (Baseline) | 43.57 | 50.64 | 52.96 |
> |  | 46.50 | 53.22 | 60.43 |
> |  | 46.50 | 53.68 | 62.64 |
> | LLaDA-8B-Instruct  (P-POTS+Mirror) | 50.80 | 62.02 | 66.02 |
> |  | 53.80 | 58.61 | 66.67 |
> |  | 53.00 | 60.96 | 68.62 |
>
> (In the main table reported in our paper, the “±” values are standard deviations over the three runs. For example, for OpenScience under the standard method we compute
> $\sqrt{((43.57-m)^2 + (46.50-m)^2 + (46.50-m)^2 )/2}$, where $m$ is the mean.)
>
> **From the table, across three runs, even the best performance of the standard method often remains well below worst performance of PPOTS+MIRROR:**
>
> a) OpenScience: best standard = 46.50 vs. worst P-POTS+MIRROR = 50.0.
> b) GSM8K: best standard = 53.68 vs. worst ours = 58.61.
> c) HiTab: best standard = 62.64 vs. worst ours = 66.02.
> d) text-to-image-2m: best standard = 34.28 vs. worst ours = 34.10.
>
> **Beyond improving average performance, our method also reduces run-to-run variability to nearly the same level as ARMs (PPOTS+MIRROR VS Llama-3.1-8B). Practically, this means each training run produces a stable and stronger model, instead of requiring multiple expensive training to “find a good run” under the standard method -- and even then, that best-case baseline is often still below our worst-case result.**
>
> ---
>
> ## 3) General and Knowledge-Extensive QA tasks Applicability
> OpenScience is a multi-domain knowledge-extensive dataset spanning STEM, law, economics, and humanities. A representative example is:
>
> {
>
> “input”: “Which of the following technological advancements is **not** typically associated with the Second Industrial Revolution?  A: The light bulb  B: The telephone  C: The assembly line  D: The steam engine  E: The automobile  F: The radio  G: The airplane  H: The electric motor  I: The phonograph  J: The refrigerator”,
>
> “output”: …
> }
>
> As shown in our main table, our method performs on OpenScience just as strongly as on other reasoning-focused benchmarks.
>
> ---
>
> In conclusion, we are grateful for your constructive feedback, and we hope these clarifications address your concerns. We would like to emphasize that our method **consistently outperforms across runs**, both by markedly improved average performance and by reduced run-to-run variability to nearly the ARM level. This is crucial, because the uniquely high training variance in MDMs leads to inherently noisier gradient updates, which makes training for MDMs much harder than that of ARMs.

---

> ### Author Response · Authors · 2025-11-28
>
> Dear Reviewer WDaB,
>
> We hope you are doing well. We are writing to gently follow up our previous comment, where we clarified a potential misunderstanding about our experimental setup and results. We really appreciate the time you have already devoted to our paper and would be grateful if you could take a quick look at our clarification when convenient.
>
> Best regards,
>
> The authors

---

### Official Review · Reviewer_ag9a · 2025-10-31

**Soundness:** 2
**Presentation:** 3
**Contribution:** 3
**Rating:** 4
**Confidence:** 2

**Summary:**

The paper analyzes training instability in masked diffusion models by decomposing the loss variance into three parts: (A) randomness from the mask pattern, (B) randomness from the mask ratio/timestep, and (C) data variance. To reduce these sources, it introduces P-POTS (a timestep sampler), Mirror (a complementary mask), and various other techniques. Across multiple benchmarks, the proposed approaches stabilize training and improve final results.

**Strengths:**

Overall, I think the paper studies an important problem. The proposed fixes P-Pot and Mirror are clearly argued and have demonstrated practical usefulness in terms of accuracy and training stability.

**Weaknesses:**

1. The loss-variance decomposition is insightful, but I believe for training stability it would be more insightful to analyze gradient variance. It would be nice to see how reductions in the proposed loss variances translate to reduced gradient variances and more stable optimization.
2. MIRROR roughly doubles the cost on some benchmarks compared to the baselines, which is quite expensive. Would MIRROR still be the best choice under a fixed time budget, which is a more practical scenario?
3. I am not familiar with MDMs and thus cannot comment on the commonly reported numbers. But for ARMs, the Qwen-2.5-7B-Instruct and Qwen-3-8B numbers appear below the commonly reported results. Could the authors clarify their evaluation approachs?

**Questions:**

The variance decomposition is not unique. It depends on the conditioning order when you iteratively apply the law of total variance. Could the authors discuss how your conclusions change under alternative decompositions, and whether those alternative decompositions could lead to other interesting approaches?

---

> ### Author Response · Authors · 2025-11-22
>
> Thank you for your thoughtful review. We are glad you found our study of training instability in MDMs important, and we especially appreciate your suggestions on strengthening the theoretical justification. Some of the points you raised were part of our internal discussion, but we did not include them in the main paper due to the tight page limit and already high formula density. We will add them in the revision, and please let us address your concerns point by point below.
>
> ---
>
> ## 1) Loss variance vs. gradient variance
> We agree that gradient variance is the quantity most directly tied to optimization stability. In our preliminary exploration, we examined both loss variance and gradient variance. We finally turned to loss variance for three reasons:
> 1. **Optimizer-agnostic improvement (a property that gradient variance does not enjoy).**
> In practice, models are optimized with various optimizers, such as AdamW, Adam, AdaFactor, SGD, and $\mathrm{Var}(g)$ is tightly coupled to the specific optimizer choice. Adaptive methods (Adam/AdamW/AdaFactor) rescale gradient coordinates, so the “raw” gradient variance no longer faithfully reflects the variance of actual parameter updates. In contrast, minimizing Var(l) directly improves the SNR ratio of the MC estimate of what training ultimately estimates, $E_{x_0, t, x_t}(l_\theta)$.
> 2. **Multi-parameter conflict (gradient is not a scalar).**
> For an 8B model, $g\in\mathbb{R}^{8\text{B}}$. Thus, minimizing “gradient variance” requires scalarizing it (e.g., using $\|g\|_2^2$), which is not theoretically neutral: different parameters may prefer different sampling distributions.  Any scalarization therefore introducex extra design choices and breaks the clean optimality claim. In contrast, the loss is already a unified scalar with a clear interpretation as the objective whose expectation we estimate, so its variance is unambiguous.
> 3. **Loss variance is the only quantity the t-sampler can reliably control.**
> The gradient variance is
>    $
>    \mathrm{Var} \left(\nabla_\theta \hat{\mathcal{L}}\right)
>    =\mathrm{Var} \left(\frac{1}{N}\sum_{i=1}^N\nabla_\theta l_\theta(x_0^{(i)},t^{(i)},x_t^{(i)})\right).
>    $
>    By first-order linearization,
>    $
>    \mathrm{Var}(\nabla l)\approx J_\theta\mathrm{Var}(l) J_\theta^\top,
>    $
>    where $J_\theta$ is the Jacobian of the model w.r.t. parameters.
> This highlights that part of gradient variance comes from **model sensitivity (the Jacobian)**, which a $t$-sampler cannot affect. The only reliably controllable component is $\mathrm{Var}(l)$. Therefore, reducing loss variance is a controllable way to reducing gradient variance (holding $J_\theta$ fixed over short horizons).
>
> ---
>
> ## 2) How your conclusions change under alternative decompositions, and whether those alternative decompositions could lead to other interesting approaches.
>
> **Conclusion: no; the theoretically optimal sampler $p^\*$ we derived is unique, because it depends only on the conditional second moment given $t$. Hence the derivation is independent of conditioning order. Different orders do yield different decompositions, but the order we use ($x_0 \to t \to x_t$) is the only interpretable one and with a clean ARM comparison.**
>
> Indeed, different conditioning orders yield different decompositions. For example:
>
> $$x_0 \to t \to x_t$$
>
> $$x_0 \to x_t \to t$$
>
> However, only the first order (the order used in our paper) corresponds to how diffusion training is executed: we sample $x_0$, sample $t$, then generate $x_t$ by masking tokens i.i.d. with rate $t$. Under this order, the decomposed terms
> - $A_1$ captures mask-pattern noise (variance over which tokens are masked given $(x_0,t)$);
> - $B_1$ captures mask-ratio noise (variance due to different $t$ values);
> - $C_1$ captures intrinsic data difficulty (variance over $x_0$).
>
> The $x_0 \to x_t \to t$ order is mathematically valid but hard to interpret in diffusion training, since $x_t$ is not defined until we choose $t$. Moreover, the ARM decomposition $\mathrm{Var}_{x_0}(l)$ naturally aligns with the outermost $x_0$ term in the first order (i.e., $C_1$), which is not true for other orders.
>
> Also, since the optimal $t$-sampler depends only on the conditional second moment, so its form is irrelevant to conditioning orders.
>
> (other concerns are addressed in the next comment)

---

> ### Author Response · Authors · 2025-11-22
>
> ## MIRROR cost and why it is cost-effective
> We agree that MIRROR roughly doubles the per-step cost. However, baseline MDM training variance is so high that a single run is not sufficient to obtain a reliable and high-performing model, so repeated training may be needed (see table below). In this case, it exceeds MIRROR's doubled cost. To demonstrate this, in our paper, we repeat **training (not inference)** for three times with different seeds (42, 731, 20231), holding model/data/hyperparameters fixed, and evaluate trained models with greedy decoding so that differences across runs only come from training, not inference. The results are summarized as follows:
>
> | Model (Method) | OpenScience | GSM8K | HiTab |
> |---|---:|---:|---:|
> | **ARMs** ||||
> | Qwen2.5-7B-Instruct | 46.30 | 54.66 | 45.74 |
> | Qwen3-8B-Base | 51.70 | 70.58 | 48.84 |
> | Llama-3.1-8B-Instruct | 44.95 | 47.60 | 65.02 |
> |  | 47.07 | 46.40 | 66.28 |
> |  | 47.76 | 49.10 | 64.88 |
> | **MDMs** ||||
> | LLaDA-8B-Instruct (Baseline) | 43.57 | 50.64 | 52.96 |
> |  | 46.50 | 53.22 | 60.43 |
> |  | 46.50 | 53.68 | 62.64 |
> | LLaDA-8B-Instruct  (P-POTS+Mirror) | 50.80 | 62.02 | 66.02 |
> |  | 53.80 | 58.61 | 66.67 |
> |  | 53.00 | 60.96 | 68.62 |
>
>
> This table shows that although pretrained MDMs are reported to have comparable ability with strong ARMs, their uniquely high training variance poses a significant training gap and prevents them to achieve full potential. **Even with three restarts, the best baseline model remains well below our worst-case P-POTS+MIRROR run**. This supports our claim that P-POTS is not heuristic but follows directly from variance minimization and is therefore theoretically optimal among all $t$-samplers.
>
> ---
>
> ## Clarifying ARM numbers on GSM8K / HiTab / OpenScience
> - **GSM8K.** Our paper reports strict pass@1 with greedy decoding and a generation length of 512, because the training data has max response length = 444 tokens. Publicly reported Qwen GSM8K numbers are often obtained under less strict pass@8 or pass@32 settings with 4k-8k generation length. Our goal here is controlled comparison of training variance, hence we use a fully deterministic evaluation.
>
> - **HiTab.** HiTab is a tabular QA benchmark. To the best of our knowledge, Qwen2.5-7B-instruct and Qwen3-8b-base are not evaluated on common tabular benchmarks, except for TQA-bench, where Qwen2.5-7B-Instruct achieves 49.86% pass@1 at 8K context. this is consistent with our controlled setting.
> - **OpenScience.** We did not find previous ARM evaluations on OpenScience either. This dataset spans across STEM, law, economics, and humanities, so we believe our ARM results are reasonable.
>
> ---
>
> In conclusion, we hope this reply addresses your concerns and clarifies both our theoretical choices and empirical findings. We are happy to answer any further questions you may have.

---

> ### Author Response · Authors · 2025-11-28
>
> Dear Reviewer ag9a,
>
> We hope you are doing well. We would like to briefly follow up on our earlier comment regarding your review, where we have provided a detailed response to your theoretical concerns. We really appreciate the time you have already spent on our paper and would be grateful if you could take a quick look at our clarification when convenient.
>
> Best regards,
>
> The authors

---

### Official Review · Reviewer_oCXk · 2025-11-01

**Soundness:** 3
**Presentation:** 3
**Contribution:** 3
**Rating:** 6
**Confidence:** 3

**Summary:**

The paper introduces a theoretical and practical framework to understand and address instability in masked diffusion model (MDM) training. The authors first derive a variance decomposition framework that attributes training variance to three distinct sources: (A) masking pattern noise, (B) masking ratio noise, and (C) data noise. Based on this decomposition, they propose six variance-reduction methods. Experiments on multiple text reasoning datasets (GSM8K, HiTab, OpenScience) and a text-to-image benchmark show consistent improvements in both performance and stability. Overall, the paper is technically strong, though somewhat dense and in need of a tiny bit of clearer writing.

**Strengths:**

1. Strong theoretical foundation. The paper provides a clear and principled variance decomposition for masked diffusion model (MDM) training, unifying prior ad-hoc stabilization methods under a single theoretical framework. It then builds directly on this foundation by proposing six targeted variance-reduction techniques to mitigate the identified sources of instability.
2. Comprehensive empirical validation. The experiments cover both language and multimodal domains, demonstrating that the proposed methods are broadly effective and improve training stability across diverse settings.

**Weaknesses:**

1. Narrow comparison to ARMs. The study includes only two autoregressive baselines from the same family. Incorporating additional ARM baselines, especially models with different architectures or training paradigms, would help clarify whether the observed variance gap is a general phenomenon or specific to the chosen comparison set.
2. Limited model diversity and scaling analysis. While the empirical results are solid, they are restricted to a single MDM backbone (LLaDA-8B-Instruct). Evaluating the proposed methods across different model sizes and architectures would strengthen the claims of generality and potentially reveal whether the improvements follow any scaling trends within or across model families.
3. Writing and presentation. The exposition could be tightened to improve readability; the current density of equations and notation can make the paper feel more complex than necessary. Additionally, some figures could benefit from more informative captions, for instance, Figure 3 presents image generation results but omits the corresponding prompts.

**Questions:**

See Weakness.

---

> ### Author Response · Authors · 2025-11-22
>
> Thank you very much for your constructive review. We are glad that you found our theoretical framework sound and our empirical validation comprehensive. Below we address your concerns point by point.
>
> ---
>
> ## 1) Narrow comparison to ARMs
>
> **A central claim of this paper is that the uniquely high training variance in MDMs leads to noisy gradient updates. Hence even pretrained MDMs like LLaDA show comparable ability with strong ARMs, they often underperform ARMs by a wide margin once trained.** Our method reduces this variance and closes the training gap, enabling MDMs to match or even surpass ARMs. To validate this claim, we initially compared against what we believe are among the strongest ~8B ARMs, Qwen2.5-7B-Instruct and Qwen3-8B-Base.
>
> As you kindly pointed out, these two baselines come from the same family and may not fully represent general ARM performance. We therefore add a third ARM from a different family: meta-llama/Llama-3.1-8B-Instruct.
>
> Following the same protocol as for MDMs, we repeat **training (not inference)** with three random seeds (42, 731, 20231), keeping the model, data, and hyperparameters fixed, and using **greedy decoding at inference** so that run-to-run differences come only from training stochasticity.
>
> ### Updated results (Perf across seeds 42 / 731 / 20231)
>
> | Model (Method) | OpenScience | GSM8K | HiTab |
> |---|---:|---:|---:|
> | **ARMs** ||||
> | Qwen2.5-7B-Instruct | 46.30 | 54.66 | 45.74 |
> | Qwen3-8B-Base | 51.70 | 70.58 | 48.84 |
> | Llama-3.1-8B-Instruct | 44.95, 47.07, 47.76 | 47.60, 46.40, 49.10 | 65.02, 66.28, 64.88 |
> | **MDMs** ||||
> | LLaDA-8B-Instruct (Baseline) | 43.57, 46.50, 46.50 | 50.64, 53.22, 53.68 | 52.96, 60.43, 62.64 |
> | LLaDA-8B-Instruct  (P-POTS+Mirror) | 50.80, 53.80, 53.00 | 62.02, 58.61, 60.96 | 66.02, 66.67, 68.62 |
>
> These additional cross-family ARM comparisons reinforce our main conclusion: **our method effectively closes the post-training gap between MDMs and ARMs, allowing MDMs to reach their true potential.** With our method (PPOTS+MIRROR), the average performance is substantionally improved, with run-to-run variability reduced to near-ARM level.
>
> ## 2) Limited model diversity and scaling analysis
>
> In this paper, we experiment on two MDMs: LLaDA-8B-Instruct (language) and MMaDA-8B-MixCoT (multimodal). We agree that broader model diversity and scaling would be valuable, but we had faced substantial practical barriers when writing this paper:
>
> 1. **Limited available MDM checkpoints.** Open-sourced pretrained MDMs are far fewer than ARMs (to the best of our knowledge, fewer than 10).
>
> 2. **No public training code.** Existing MDM releases (including LLaDA and MMaDA) provide only inference code. When writing this paper, we had to re-implement the complete MDM training code from scratch (≈3k+ lines of code). This “from-scratch reproduction + verification” effort was substantial, and it made extending to many additional MDM baselines non-trivial within the scope of this work.
>
> 3. **Computational constraints.** To demonstrate consistent run-to-run performance, we trained each setting with three seeds, tripling compute cost (in total, our study required roughly 500 H100 hours for training and 330 H100 hours for inference.)
>
> We will clarify these constraints more explicitly in the revision, and we view broader scaling and additional MDM families as an important direction for future work.
>
> ---
>
> ## 3) Writing and presentation
>
> Thank you for highlighting the presentation issues. We will revise the paper to improve readability by:
>
> - moving several derivations to the appendix,
> - streamlining the main text,
> - and expanding figure captions with more implementation and experimental detail.
>
> ---
>
> In conclusion, we sincerely appreciate your suggestions on both baseline choices and presentation. They have helped us strengthen the empirical support and improve the clarity of the paper.

---

> ### Comment · Reviewer_oCXk · 2025-11-22
>
> Thank you for the detailed response, and for including an additional ARM baseline. The new results and explanations satisfactorily address my concerns, and I appreciate the clarification around the practical constraints that limited model diversity and scaling experiments. **I will be maintaining my original score.**
>
> Additionally, as the authors noted that training code for MDM models is not widely available, the reviewer encourages the authors to consider releasing their implementation (along with baselines). This would greatly benefit the community and support further research in this emerging area.

---

> > ### Author Response · Authors · 2025-11-22
> >
> > Thank you for the suggestion. We are currently cleaning up our training/evaluation codebase, and we plan to open-source the full implementation of our proposed methods in a timely manner to support reproducibility and follow-up research.

---

### Author Response · Authors · 2025-12-02
**General Response**

We sincerely thank all reviewers for their valuable time and constructive feedback. We are delighted that our work is viewed as having a strong theoretical foundation and strong empirical evidence, and as studying an important theorem.

Your comments have greatly helped us improve the clarity and quality of the paper. We have uploaded a revised PDF accordingly, and we detail what has been changed and why in a separate response. These changes are minor and made mostly to improve the clarity and presentation of our original paper. The experimental setup and our main conclusions remain unchanged.

**What was changed**

- In Section 3.2, we replaced the previously reported aggregated results with the per-run results. These per-run results were already included in the original submission in the appendix; we have now moved them into the main text so that it is more easy to see the behavior of each individual run.
- We also added results for one autoregressive models as an additional baseline. This modification is small compared to the main experiments, and the new results are fully consistent with our original conclusions.
- We added a theoretical explanation of the variance decomposition to make the intuition more explicit. The formal statements and conclusions of our analysis remain unchanged.


**Why we made these changes**

- Some reviewers found it unclear that the previous results were aggregated over three independent training-evaluation runs, and therefore could not fully assess the ability of our method. By showing the per-run results directly in the main text, we aim to make it more transparent that our method **consistently outperforms the baselines by a wide margin**, and to reduce any potential misunderstanding.

- Similarly, the added theoretical clarifications on the variance decomposition are intended to better connect the formal results with the empirical behavior observed in our experiments, without altering the underlying conclusions.

We hope this reorganization and clarification make the experimental section and theoretical analysis easier to interpret.

Best regards,
The authors

---

### Meta-Review · Area_Chair_bnVY · 2025-12-23

**Summary:**

Reviewers raised several concerns, including: (i) the need for broader empirical comparisons, (ii) the computational overhead introduced by the MIRROR method, (iii) the focus on loss-variance reduction rather than a direct analysis of gradient variance, and (iv) the absence of train-from-scratch experiments.

While some of these concerns remain partially unaddressed (eg., train-from-scratch evaluation), the proposed approach is theoretically well-grounded, demonstrates consistent and substantial empirical gains, and is validated across a diverse and meaningful set of benchmarks. The rebuttal and revisions address the major empirical and clarity-related concerns raised by the reviewers.

Based on the overall strength of the theoretical analysis, and empirical evidence in the studied scenarios, and the reviewer discussion, the AC’s assessment is that the initial reviewer scores (8, 6, 6, 4) would likely increase. The paper is therefore recommended for acceptance. BTW, please check the caption of Fig.3.

**Reviewer Concerns:**

**Concerns addressed by the rebuttal:**

* **Reviewer oCXk:**
  The concern regarding limited ARM comparisons was addressed by adding an additional cross-family baseline (Llama-3.1-8B) and clarifying the evaluation protocol.

* **Reviewer ag9a:**
  The concern about focusing on loss variance rather than gradient variance was addressed through a detailed justification emphasizing optimizer-agnosticity and the limited controllability of gradient variance via sampling. Concerns regarding the computational cost of MIRROR were addressed by arguing that reduced run-to-run variance mitigates the need for multiple training runs, leading to lower effective compute cost in practice. Clarifications were also provided regarding evaluation settings (e.g., strict pass@1 with greedy decoding).

* **Reviewer WDaB:**
  Concern about limited experiments performance was addressed. The comment on evaluations on standard QA tasks is addressed by arguing the OpenScience benchmark covers this.

* **Reviewer J4Ut:**
  Questions regarding applicability to other MDM variants (e.g., Dream and Block Diffusion) were addressed conceptually, with explanations of how the proposed methods extend to these settings.

**Concerns still outstanding:**

* **Train-from-scratch evaluation** (Reviewer J4Ut):
  While acknowledged as an important direction, full train-from-scratch experiments were not conducted due to computational constraints and remain future work.

* **Broader model and benchmark diversity** (Reviewers oCXk, WDaB):
  Although the authors provided reasonable justification regarding the limited availability of public training code and pretrained checkpoints, broader coverage across additional MDM families and tasks remains an open direction.

**Reviewer Scores:**

Reviewer **oCXk** would likely maintain their original score, as their main concerns (e.g., broader ARM comparisons) were directly addressed in the rebuttal, and the reviewer explicitly indicated satisfaction with the responses. Reviewer **ag9a** would likely increase their score, as the rebuttal provided substantive clarifications on the choice of loss-variance analysis over gradient-variance analysis and justified the computational cost of MIRROR under realistic training scenarios. Reviewers **WDaB** and **J4Ut** would likely keep their (already positive) scores, as their main concerns were addressed. The comment on train-from-scratch evaluation was also reasonably framed as future work and does not materially affect the overall strength of the contribution.

---

### Decision · Program_Chairs · 2026-01-26

Accept (Poster)